# Modelling dataset bias in machine-learned theories of economic decision-making

Tobias Thomas [1,2] ✉, Dominik Straub [1], Fabian Tatai [1], Megan Shene[1], Tümer Tosik [1], Kristian Kersting[2,3] & Constantin A. Rothkopf[1,2]

Normative and descriptive models have long vied to explain and predict human risky choices, such as those between goods or gambles. A recent study reported the discovery of a new, more accurate model of human decision-making by training neural networks on a new online large-scale dataset, choices13k. Here we systematically analyse the relationships between several models and datasets using machine-learning methods and find evidence for dataset bias. Because participants' choices in stochastically dominated gambles were consistently skewed towards equipreference in the choices13k dataset, we hypothesized that this reflected increased decision noise. Indeed, a probabilistic generative model adding structured decision noise to a neural network trained on data from a laboratory study transferred best, that is, outperformed all models apart from those trained on choices13k. We conclude that a careful combination of theory and data analysis is still required to understand the complex interactions of machine-learning models and data of human risky choices.

Human choices between goods or gambles have long been known not to maximize expected gains[1,2]. Therefore, predicting and explaining how and why humans make decisions has been a major goal in psychology, economics, cognitive science and neuroscience[3–6]. While normative models aim to explain why people should make certain decisions, descriptive models have tried to capture how people actually decide. In this vein, classic economic theory has devised axiomatic approaches to decision-making[7,8] with the goal of explaining human choices by starting from first principles and deriving mathematically how one should decide. On the other hand, the widespread observation that humans regularly violate these axioms and accordingly systematically deviate from the predictions of these normative models has led to the development of descriptive decision-making models[9] and the inclusion of cognitive factors into models within behavioural economics[10]. Such descriptive models do not necessarily explain why people adopt particular policies, but they can aid in predicting them.

Over the past decade, advances in machine learning, particularly data-driven methods involving neural networks (NNs), have led to remarkable advances in the natural sciences, for example in physics[11],

engineering[12] and biology[13]. This trend has also extended to developing NN models of human decision-making and training them on newly collected datasets[14–17]. The promise of these efforts is not only to obtain more accurate descriptive models of human decisions, but to explain human decisions and advance the theory of human decision-making. Although Peterson et al.[17] caution that 'Human ingenuity will also be required for potentially translating this descriptive theory into normative and process models', their paper has spurred excitement about the potential automation of theory development. As recently put in a commentary: 'Instead of relying on the intuitions and (potentially limited) intellect of human researchers, the task of theory generation can be outsourced to powerful machine-learning algorithms'[18].

However, theory, models and data are related in intricate ways[19–23]. While work in epistemology and philosophy of science has contended that theory is a prerequisite to any data collection[24,25], analysing data that have been collected requires taking into account fundamental computational properties relating models and data. Specifically, when training NNs to model human decision-making, it is important to (1) obtain representative data of choice behaviour for training and testing

[1]Centre for Cognitive Science and Institute of Psychology, Technical University of Darmstadt, Darmstadt, Germany. [2]Hessian Center for Artificial Intelligence, Darmstadt, Germany. [3]Centre for Cognitive Science and Computer Science Department, Technical University of Darmstadt, Darmstadt, Germany. ✉e-mail: tobias.thomas@tu-darmstadt.de

models and to (2) select models that balance interpretability with expressiveness. Complicating matters, additionally, it is necessary to (3) investigate how the datasets and models interact. As an example, although a more complex model should on average always fit a dataset better than a less complex model, a small dataset generated from a complex model may be explained better by a simpler, less expressive model, even if it is not the model that generated the data, while a larger dataset generated from a complex model may be better explained by the complex model that actually generated the data.

First, it could be that the training set is too small for the model at hand, because the more flexible a model, the more it becomes fundamentally important to avoid, for example, overfitting and bias–variance trade-offs[26]. Overfitting occurs when the machine-learning model captures the regularities of data it has been trained on arbitrarily closely, but its predictions' accuracy does not transfer well to unseen test data, that is, it fits the noise. Indeed, NN models might be heavily over-parameterized, although such heavily over-parametrized NNs may sometimes exhibit good generalization to unseen test data. This phenomenon is known as double descent and is far from being fully understood, with some initial theoretical work relating it to the dataset as well as the model at hand[27–29]. However, such possible interactions between datasets and models need to be tested empirically. A second case relates to NNs that have been optimized using training and testing sets, but nevertheless show idiosyncratic generalization to new situations. Common examples in vision include changes to images that are invisible to the human eye but result in their misclassification[30]. A third case relates to the trained NN showing high predictive performance on a dataset but for the wrong reasons, for example, by picking up on spurious correlations in the training data[31–33]. Finally, another case arises when the trained model does not transfer between datasets simply because two datasets have different properties, such as their data distributions. This so-called dataset bias, which includes the prominent selection bias, is pervasive in modern machine learning and has been described repeatedly, including in NNs trained on object recognition tasks[34].

In two notable studies aimed at deriving new theoretical insights from using NNs to predict human choices between gambles, Bourgin et al.[16] and Peterson et al.[17] addressed the above points by (1) collecting a new dataset whose size exceeds by far all previous collections of human decisions under risk. The authors obtained this impressive dataset, called choices13k, which contains human choices on over 13,000 different choice problems, through Amazon Mechanical Turk (AMT). This is a considerable achievement, given that the size of previous datasets, for example, the choice prediction competition 2015 (CPC15) dataset collected for a choice modelling competition contains choices by 446 participants in laboratory experiments at the Hebrew University of Jerusalem (HUJI) and the Technion for 150 different choice problems[14]. Peterson et al.[17] addressed point (2) by using cross-validation in training the models and by employing a succession of NN models of increasing expressiveness, implementing various constraints. These constraints were carefully selected to help the interpretability of the NNs' behaviour in psychological terms. Progressively lifting these constraints allowed NNs to incorporate 'contextual effects' ranging from violations of independence and transitivity axioms to complex interactions of transformations of probabilities, outcomes and information across gambles. Peterson et al.[17] developed this methodology of machine-learned theories to find interpretable differentiable models of human decisions instead of hard to interpret, complex NN functions. However, perhaps not surprisingly, Peterson et al. reported that the most flexible, 'fully unconstrained' NN with the capacity to express arbitrary mappings of probability weightings and utility weightings with contextual effects best-fitted human decisions on the new dataset, choices13k.

In this Article, we take a step back and investigate the relationship between datasets, models and theory by first investigating possible interactions between models and datasets. To this end, we systematically trained several machine-learning models on choice datasets from three different studies, CPC15[14], choice prediction competition 2018 (CPC18)[35] and choices13k[17], and analysed the resulting models' differences in predictions; see Fig. 1 for a schematic. First, we find clear signatures for dataset bias by applying transfer testing. To investigate the possible source of this dataset bias, we asked which features of gambles are predictive of the difference in predictions between models trained either on CPC15 or choices13k. Using linear models and a popular technique in explainable artificial intelligence (XAI), feature importance weights[36], shows that features derived from the psychology and behavioural economics literature allow for capturing the deviations better than base features of gambles. Particularly three features, which all relate to the degree to which one gamble is expected to yield a higher payoff than an alternative, were predictive of the difference in the predictions of NN models trained separately on the two datasets. Because these three features consistently predicted less extreme proportions of choices for the NN trained on choices13k compared with the NN trained on CPC15, and because previous research suggests that behavioural data obtained on AMT is more variable compared with laboratory settings[37], we hypothesized that the source of the difference between datasets is decision noise. Indeed, a hybrid model involving a generative Bayesian network modelling a proportion of subjects as guessing and the remaining subjects generating choices according to the NN trained on CPC15, with added decision noise in log-odds space, accounts for more than half of the discrepancy between datasets and transferred best from CPC15 to choices13k. Taken together, this clarifies that size of datasets alone is not a sufficient warrant for devising general theories of human decision-making and that the context of data collection may be included in the modelling, as it has a notable impact on decisions. Finally, combining machine learning, data analysis and theory-driven reasoning are currently still helpful in predicting and understanding human choices in economic decision-making tasks and may guide in devising future research questions.

## Results
### Choice data
First, we systematically investigated the interplay between decision datasets and machine-learning models. These analyses included checking for potential dataset bias and overfitting or idiosyncratic generalization. Accordingly, we reasoned that if a model trained on choices13k, the largest dataset so far, captures human decision-making well, its predictions should consequently generalize to other datasets collected previously, particularly because these previously collected datasets are smaller. Similarly, if different models trained on the same dataset generalize comparatively well, overfitting and idiosyncratic generalization should be considered less likely. Therefore, we trained previously employed machine-learning models on the CPC15 and choices13k training datasets and quantified their performance on the training and test data of the respective other dataset (Table 1). This is a common way to quantify how well the models generalize from one dataset to the other and is known as transfer testing in machine learning[38]. Additionally we tested all models on the train and test sets of the data from CPC18 (ref. 35) with compatible format. Note that the training set of CPC18 contains all of CPC15, not only including gambles, but also behavioural data.

### Choice models and training
We used a total of five different models, including three classical machine-learning methods that performed among the best in the CPC challenges: the psychological model Best Estimation And Sampling Tools (BEAST)[14], random forest[39] and support vector machine (SVM)[15]. Additionally, we trained two different NN architectures, specifically the NN presented in Bourgin et al.[16] and the most expressive, context-dependent NN presented in Peterson et al.[17]. Expressiveness here relates to the complexity of functions the model can potentially

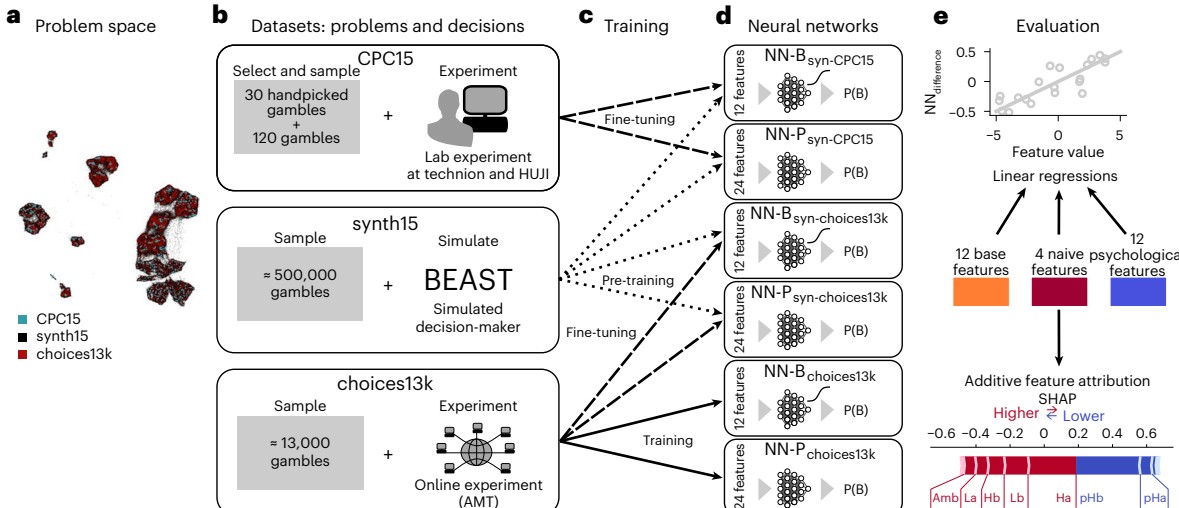

**Fig. 1 | Schematic of analyses of the relationship between datasets and models. a**, Because all pairs of gambles considered in this study can be parameterized in one common way, the decision problems' features can be used to compute a two-dimensional embedding (uniform manifold approximation and projection), representing the problem space. Each dot corresponds to a decision problem consisting of two gambles and the colours indicate the dataset of origin. **b**, Pairs of gambles from this problem space, together with the proportion of choices, constitute the datasets: the CPC15 dataset with human decisions from a laboratory study, the choices13k dataset with human decisions from a large-scale online experiment and a much larger synthetic dataset (synth15) generated by predictions from the psychological model BEAST. **d**, We trained six different NNs on the basis of two architectures; NN$_{Bourgin}$(NN-B)), based on Bourgin et al.[16] and NN$_{Peterson}$(NN-P), based on Peterson et al.[17]. **c**, The target of training was the proportion of trials in which gamble B was chosen, averaged over all human participants and 5 trials (P(B)) from either CPC15 or choices13k. However, because of the small size of the CPC15 dataset, we first pre-trained on synth15 and then fine-tuned on CPC15. To test for dataset bias, we also pre-trained some NNs on synth15 and then fine-tuned on choices13k. We can now investigate the relationship between models and datasets by comparing predictions of NNs on decision problems. Because all pairs of gambles reside in the same problem space but the overlap in decision problems across datasets is small, we compute the difference in predictions between any two models on problems sampled from the problem space. **e**, Subsequently, we investigated the source of the differences in predictions between different combinations of models and datasets. First, we use linear regressions (top), relating individual or sets of features of the gambles to the difference in model predictions. Second, we use SHAP[36], an XAI method, which returns linear additive feature importance values for each gamble (bottom).

**Table 1 | Transfer testing between models trained on CPC15 and choices13k**

| Training set | Model | Testing on: (MSE×100) | | | | | |
| --- | --- | --- | --- | --- | --- | --- | --- |
| | | CPC15 | | choices13k | | CPC18 | |
| | | Train | Test | Train | Test | Train | Test |
| CPC15 | BEAST | 1.34 | 0.98 | 2.47 | 2.53 | 1.01 | 0.76 |
| | Random forest† | 0.32 | 1.10 | 1.88 | 1.76 | 0.82 | 1.62 |
| | SVM† | 0.45 | 1.65 | 2.39 | 2.24 | 1.07 | 2.47 |
| | NN$_{Bourgin,Prior}$ (NN$_{CPC15}$) | 0.28 | 0.53 | 2.69 | 2.77 | 0.50 | 0.64 |
| | NN$_{Bourgin,Prior}$ + decision noise | 1.24 | 1.43 | 1.49 | 1.53 | 1.44 | 1.48 |
| choices13k | Random forest† | 1.75 | 1.44 | 0.58 | 1.03 | 1.54 | 1.70 |
| | SVM† | 2.56 | 1.97 | 0.73 | 1.00 | 1.72 | 1.97 |
| | NN$_{Bourgin,Prior}$ | 1.92 | 1.38 | 0.90 | 1.00 | 1.73 | 1.56 |
| | NN$_{Bourgin}$ (NN$_{choices13k}$) | 2.66 | 1.94 | 1.25 | 1.30 | 2.50 | 2.38 |
| | NN$_{Peterson,Prior}$ | 2.13 | 1.83 | 1.27 | 1.34 | 1.96 | 1.95 |
| | NN$_{Peterson}$ | 2.80 | 2.01 | 1.31 | 1.33 | 2.26 | 2.24 |

Generalization of different machine-learning models trained on CPC15 or choices13k and tested on CPC15, choices13k and CPC18 in terms of the MSE×100. Models trained on the much larger AMT dataset (choices13k) perform generally worse on the laboratory datasets (CPC15 and CPC18) and vice versa, giving a first indication of dataset bias. CPC18 was reduced to the subset of gambles that match the format of CPC15 and choices13k. The dagger (†) marks models that additionally use naive and psychological features as input, as opposed to only the basic features describing gambles. Note that the hybrid model adding structured decision noise to the NN trained on CPC15 transfers best to choices13k. The model with decision noise used the posterior mean of inferred parameters. Details for the decision noise are given in the 'Theory-driven modelling of the cause of dataset bias' section in Methods.

represent and learn. For example, increasing the number of neurons or hidden layers allows a NN to learn more complex functions. To investigate the influence of pre-training, we trained both NN architectures in three different ways: first by pre-training on the synth15 dataset and then fine-tuning on CPC15, second by pre-training on the synth15 dataset and then fine-tuning on choices13k, and third by training on choices13k without any pre-training. Pre-training refers to the process of training a model on another dataset or task before the actual training

process on the dataset of interest. Fine-tuning is the process of training an already pre-trained model on the target dataset. The synth15 dataset used for pre-training was originally generated by sampling a large number of gambles from the problem space that CPC15 also used and using BEAST predictions as targets[16]. The rationale for generating the synth15 dataset has been that, because of the small size of the CPC15 dataset, training a NN on CPC15 alone quickly leads to severe overfitting. Thus, pre-training on synth15, which contains synthetic data from the psychological model BEAST emulating decisions in CPC15, alleviates overfitting. Pre-training with synth15 was termed using a 'cognitive model prior' in Bourgin et al.[16]. Accordingly, we do not report results of models that were exclusively trained on CPC15 as our experiments confirmed previous findings[16,17] of overfitting on the CPC15 training data and thus no generalization to any other datasets, which would add no further insights to the current study. Additionally, we excluded the NN model based on Peterson et al. pre-trained on synth15 and fine-tuned on CPC15, because it also overfitted the training data and therefore did not provide any valuable insights.

## Establishing dataset bias

As expected, the NN trained on CPC15 performs best on the CPC15 training (mean squared error (MSE) × 100: 0.28) and test set (MSE × 100: 0.53), but generalization to the larger dataset choices13k (MSE × 100 train: 2.69, test: 2.77) is relatively poor. Performances for all models are given in Table 1. Similarly, among the models trained on choices13k, the NNs show the smallest error on the choices13k test set. The models that were additionally pre-trained on synth15 before being fine-tuned on choices13k have a slightly better generalization to CPC15 (mean increase in MSE × 100; train: 0.71 and test: 0.37) and CPC18 (mean increase in MSE × 100; train: 0.54 and test: 0.56), but perform very similarly on choices13k (mean increase in MSE × 100; train: 0.20 and test: 0.15). These results are reassuring and suggest that the NNs have learned to generalize well from the choices13k training set to the choices13k test set, that is, overfitting is not likely. However, when testing for generalization from the choices13k dataset to CPC15, performance is not consistently better but instead even worse than the transfer by the random forest model. All models, regardless of their complexity, perform much better on their respective training and test sets than on any other dataset, including the SVM and random forest (difference between the test set of the dataset trained on and both other test sets in MSE × 100; mean −0.73 and standard deviation (s.d.) of 0.51). But the expectation is that training on substantially larger datasets should transfer well to smaller datasets of the same domain, that is, if those are drawn from the same or similar data distribution. Therefore, this result has the classic signature of dataset bias, leading to the conclusion that human behaviour differed consistently and systematically between the laboratory datasets CPC15 and CPC18 and the large-scale online dataset choices13k.

## Data-driven analysis of dataset bias

Having established that participants' choice behaviour differed between the CPC15 and the choices13k datasets, the question arises how to better understand the reason for this difference. A first approach is to ask, whether particular features of the gambles are predictive of subjects' different behaviour across the two datasets. Since the gambles in choices13k are not a superset of those in CPC15, there is no set of gambles for which participants' behaviour from both studies can be compared directly. But, since the above analyses suggested that the NNs trained individually on the two datasets to predict human behaviour very accurately on their respective dataset, we can compare the predictions of two models trained individually on the two datasets with each other. For the choices13k dataset we use the NN based on Bourgin et al.[16], which was trained only on choices13k (NN$_{Bourgin}$). Additionally, analyses for the other three NN models, trained on choices13k, are reported in Extended Data Figs. 1 and 2, as well as Extended Data Table 1. For the sake

**Table 2 | Relationship between the difference in NN predictions (NN$_{difference}$) and gambles' features**

| Features | MSE | $R^2$ |
|---|---|---|
| Base | 0.0220 | 0.1186 |
| Base + naive | 0.0126 | 0.4932 |
| Base + naive + psych. | 0.0106 | 0.5760 |
| Base + naive + psych. + HOSD | 0.0103 | 0.5876 |

Linear regression between different sets of features and the difference between the NN fine-tuned on CPC15 and on choices13k. More than half of the remaining MSE and over 50% of the variance can be explained by adding naive and psychological (psych.) features. The regression was calculated on the choices13k dataset. HOSD stands for higher orders of stochastic dominance and includes second and third order.

of simplicity, these two NN models will be called NN$_{choices13k}$ and NN$_{CPC15}$, and the difference in predictions on a single gamble between these two models will be called NN$_{difference}$. Thus, we can ask, which features of gambles lead to strong deviations in predicted behaviour between NNs. To this end, we used the choices13k dataset, since it contains many more gambles than the CPC15 dataset. First, to identify which group of features explains the most variance of the difference between both NNs predictions, we used linear regressions on three different sets of features to predict the discrepancies between the two models: (1) using only the descriptive features of gambles that were used to train the NNs, (2) additionally also using the naive features, and (3) finally including the psychological features[15].

Naive and psychological features were introduced by Plonsky et al.[15] with the goal of predicting human behaviour with common machine-learning models and handcrafted features. As the names suggest, these features are either naively related to parameters of the gambles, such as the difference in expected value (diffEV) and the difference in s.d. (diffSDs) of expected values, or employ concepts from the psychological literature. Such psychological features have been developed on the basis of experimental evidence, demonstrating that human behaviour is driven by these factors, such as stochastic dominance and the probability for a gamble to generate a higher outcome (pBbet_Unbiased). This distinction thus relates to psychological theory and empirical research that has established that in certain situations peoples' choices are not only driven by the expected value of a gamble but instead by how likely it is that one gamble leads to a better outcome, independent of how much better[40]. Extended Data Table 2 lists all features with a brief explanation, for details regarding the definitions and the respective background literature, as seen in Plonsky et al.[15].

The respective MSE of the linear regressions as well as the $R^2$ values are presented in Table 2. The baseline MSE, that is, predicting the mean difference between the two NNs, is 0.0249. The table shows that the basic gamble features are unable to explain much of the difference between the two NNs in a linear regression. By contrast, including the naive and the psychological features helps to explain much more of the variance in the difference of predicted choice data, as evidenced by the reduction in half of the MSE. Thus, some of the naive and the psychological features of individual gambles are indeed capturing the difference in the predictions between NN$_{CPC15}$ and NN$_{choices13k}$. This suggests that the behaviour of subjects whose data went into the two datasets can at least in part be distinguished according to naive and psychological features of gambles.

Having established that these groups of gambles' features capture the difference between the predictions of the NNs trained on CPC15 versus choices13k, one can investigate how individual features are related to this difference between models. Therefore, we calculated the correlation between single feature values and NN$_{difference}$ for all basic, naive and psychological features. Plots for all features are provided in Extended Data Fig. 3 and a summary plot with the correlation values for all individual features are given in Fig. 2. This evaluation shows

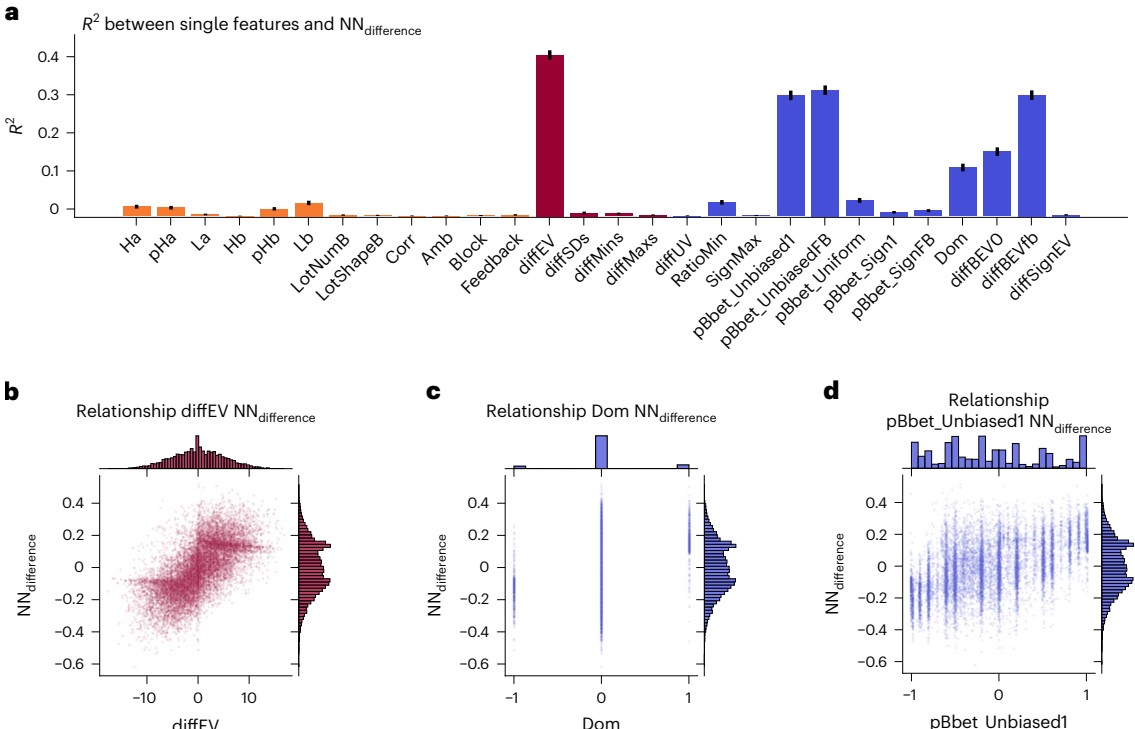

**Fig. 2 | Relationship between single features of gambles and difference in NN predictions ($NN_{difference}$). a**, $R^2$ values between single feature values and the difference between the two NN's predictions. The plot highlights that all basic features are uncorrelated with the $NN_{difference}$, while one naive and several psychological features have comparatively quite high correlations with the difference in predictions. The colour of the bar indicates the type of feature; orange is basic, red is naive and blue is psychological. The error bars indicate the 95% confidence interval, which are based on the sample size of choices13k

($N = 14,568$ gambles). **b–d**, Relationship between the diffEV (**b**), stochastic dominance (Dom) (**c**), and the probability that gamble B has a higher outcome, pBbet_UnbiasedFB (**d**) and the difference in NN predictions. These plots highlight the structure of some of the highly correlated individual features. The plot shows every gamble from the choices13k dataset as a single dot, where the $x$ axis represents the respective feature and the $y$ axis the difference between the two NNs' prediction. Additionally, marginal distributions are displayed.

that the magnitude of $R^2$ values lies between 0.001 and 0.025 for basic features, between 0.003 and 0.424 in magnitude for naive features, and between 0.001 and 0.331 for psychological features. Thus, some of the naive and psychological features are individually predictive of the difference in predictions between the two NNs, particularly the naive feature of the difference between expected value of the gambles (diffEV) and the psychological features of the probability of gamble B generating a higher outcome without (pBbet_Unbiased1) and with feedback (pBbet_UnbiasedFB), first order stochastic dominance, the estimator for the difference in expected values of gambles if gamble B is ambiguous without (diffBEV0) and with feedback (diffBEVfb). Note that all these features directly relate to the degree to which one gamble is expected to yield a higher payoff than an alternative. However, many of the psychological features that are individually predictive of the difference in the predictions of the two NN's decisions are highly correlated among each other, and thus describe related quantities. Due to this multicollinearity, features account for the same portion of variance in the difference of decisions between the two models. For details regarding the correlation between different feature values, see the feature correlation matrix in Extended Data Fig. 4.

**Theory-driven identification of dataset bias**

The analyses of the previous section showed that the difference between $NN_{CPC15}$ and $NN_{choices13k}$ are related to how much better one option is on average relative to the other. One of the most predictive single features for this difference with a correlation coefficient of 0.355 was dominance, which describes whether one gamble stochastically dominates the other one. Gamble A stochastically dominates gamble

B if, for any outcome $X$, gamble A has a higher probability than B of yielding at least $X$. An example is the gamble between option A of obtaining US$12, US$14 or US$96 with probabilities 0.05, 0.05 and 0.9, respectively, and option B of obtaining US$12, US$90 or US$96 with probabilities 0.1, 0.05, and 0.85, respectively. In this case, option A stochastically dominates option B, because the probabilities of winning US$12 or more, US$14 or more, US$90 or more and US$96 or more are 1.00, 0.95, 0.9 and 0.9 for lottery A and 1.00, 0.9, 0.9 and 0.85 for lottery B, respectively. Human behaviour in response to gambles with stochastic dominance has been studied extensively in economics, psychology and business, see ref. 41 for an example of a recent review, and therefore the predictions of the two NNs can be directly compared with previous behavioural data. Accordingly, we investigated how the predictions of $NN_{CPC15}$ and $NN_{choices13k}$ relate to the actual human choices in the respective dataset with regard to stochastic dominance of gambles. The comparison between the NN predictions and the human response rates are provided in Fig. 3a for the CPC15 dataset and in Fig. 3b for the choices13k dataset with stochastic dominance indicated by colour.

While previous analyses of the CPC15 and choices13k datasets investigated how the psychological feature of first-order stochastic dominance accounts for human choices, second- and third-order stochastic dominance have been investigated in the economics literature. Second-order stochastic dominance (SOSD) describes the preference of decision-makers for less risky gambles[42,43]. Especially in economic portfolio theory, third order stochastic dominance (TOSD) is frequently used, to order gambles or investments where all decision-makers with risk averse and absolute risk decreasing utility functions agree that one gamble is preferred[43,44]. Note that lower-order stochastic dominance is

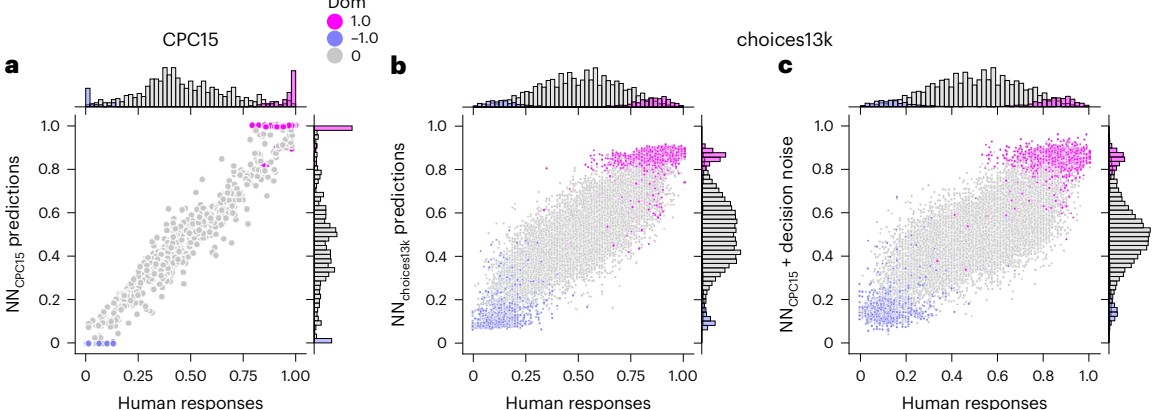

**Fig. 3 | Influence of dominance on human behaviour and NN predictions. a,b,** Comparison between human responses and their respective predictions of the NN trained on the same dataset for CPC15 (**a**) and choices13k (**b**), highlighting the different behaviour between CPC15 and choices13k, especially with respect to dominated gambles and how NNs learned this different behaviour from the two datasets. **c,** Additionally, we provide the comparison between human choices on the choices13k dataset with the NN trained on CPC15 (NN$_{CPC15}$) corrupted by

structured decision noise to show how the hybrid model captures key differences between the choice behaviour in the two datasets. This plot shows one posterior predictive sample for each gamble using the model from Extended Data Fig. 7. Each choice problem corresponds to one point, where the human response rate for gamble B is the $x$ coordinate and the model prediction is the $y$ coordinate. The colours indicate whether one of the options in the gamble stochastically dominates the other. Additionally, marginal distributions are shown.

a sufficient condition for higher order stochastic dominance (HOSD). So, if, for example, gamble A dominates B in first order, then it automatically dominates it in second and third order. Accordingly, the number of gambles with SOSD is always larger or equal to the number of gambles with first order stochastic dominance within a dataset. To test whether also higher orders of stochastic dominance may be able to explain differences across the two datasets, we calculated the features of SOSD and TOSD for all gambles. For context, in the choices13k dataset, one gamble dominates the other first order stochastically in 16%, second order in 58%, and third order in 60% of all gambles. Both of SOSD and TOSD have individually one of the highest correlations with difference in NN predictions, when being compared with the other features shown in Fig. 2a. The $R^2$ values are 0.34 (SOSD) and 0.33 (TOSD). To test whether SOSD and TOSD contain new information about the NN prediction difference, we recalculated the linear regression between feature values and difference in NN predictions from Table 2, this time including higher orders of stochastic dominance as the last set of features. The new numbers show that the two new features contain at least some additional information by slightly improving MSE and $R^2$ values.

Based on these analyses, we can draw some conclusions regarding the two datasets and stochastic dominance. First, the two datasets clearly differ strongly in terms of human behaviour in gambles with stochastic dominance. In CPC15, 95% of human responses to stochastically dominated gambles fall within the interval of choosing the dominating gamble between 0.84 and 1.00, while in choices13k, they fall within the interval 0.62 and 0.97. Thus, a higher proportion of participants in choices13k violated stochastic dominance compared with the participants in CPC15. Second, the two NNs trained on the respective dataset pick up on these different patterns of violation of stochastic dominance: the NN trained on choices13k avoids predictions of extreme choice proportions completely. Finally, two other features, which also have a high correlation with the difference in NN predictions are the diffEV and the probability that gamble B has a higher outcome value (pBbet_Unbiased). Both of these features are continuous values describing either how much (diffEV) or how certain one gamble is (pBbet_Unbiased) to yield a higher payoff. Figure 2b–d shows the relationship between the above discussed three features and the difference in NN predictions.

Taken together, the features of gambles that are able to explain most of the difference between the two NNs relate to how much 'better' one gamble is compared with the alternative, that is, the expected

payoff. Better here ranges from one gamble stochastically dominating the other, the probability that one gamble has a higher payoff than the other, to the difference between the respective expected values of gambles. For all these three features, the difference in predictions between the two NNs, reflecting the choices in the respective dataset they were trained on, are such that NN$_{CPC15}$ is much better at maximizing its return. Put differently, NN$_{choices13k}$ tends to show less extreme proportions of choices, that is, it predicts choice proportions closer to guessing.

## Automatic methods to (re)discover explanations

As NNs have become a prominent tool in machine learning but are usually obscure with respect to how the input yields an output, XAI methods have been developed attempting to provide explanations for why NN generate certain outputs. A popular class of XAI methods is called additive feature attribution methods, which forms locally linear explanations for every data point based on how important each input feature was. We used SHapley Additive exPlanations (SHAP[36]) to generate these feature importance values for the difference of the two NNs. For each data point, SHAP returns one value per each input feature, assigning an importance value to the feature for predicting that particular data point. We used SHAP to determine how much and in which direction each feature influenced the difference between both NNs. A common use case for locally linear explanation methods like SHAP is to interpret the feature importance in the original input space, for example, by visualizing them as an image. For the input features of the gambles in the CPC15 and choices13k datasets, there is no such straightforward and intuitive way of interpreting the feature importance. For this reason, and since choices13k has more than 13,000 gambles, our goal was to find structure in the SHAP values. Examples of how feature values and their respective SHAP values interact are shown in Fig. 4b–d and for all features in Extended Data Fig. 5. These plots suggest that features were typically more important in leading to a difference between both NNs when their respective magnitudes were large and that the six-base features (Ha, pHa, La, Hb, pHb and Lb) describing the gambles had the highest influence.

In the previous section, we have established that the difference in NN predictions could be partly explained using the naive and psychological features. We hypothesized that the locally linear explanations generated by SHAP might 'rediscover' the naive and psychological features. To this end, we tested whether these features could be linearly predicted by the SHAP values. The proportion of variance explained

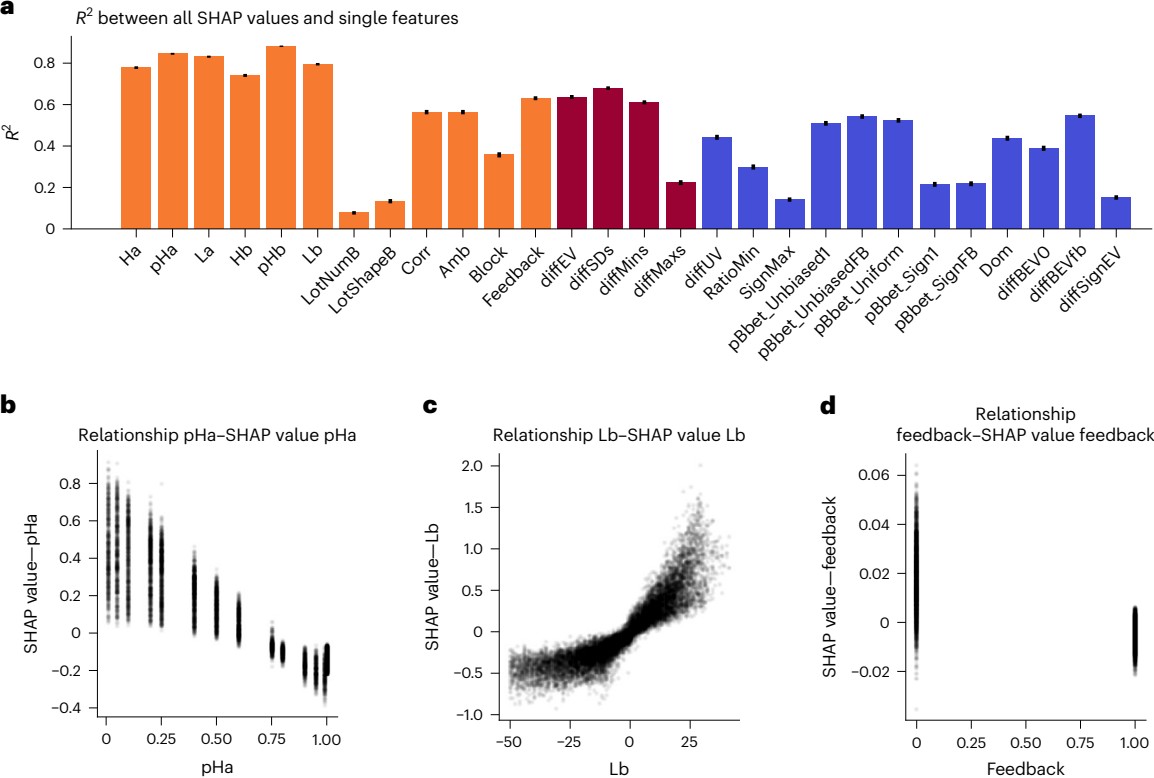

**Fig. 4 | Relationship between SHAP values and other features. a**, $R^2$ values from linear regressions between all SHAP values and single features. These linear regression results show that SHAP values are unable to explain the naive and psychological features that were especially predictive for the $NN_{difference}$, or at least not in a straightforward way. The colour of each bar indicates the type of feature: orange for basic, red for naive and blue for psychological. Regressions were calculated on the choices13k dataset. The error bars indicate the 95%

confidence interval, which are based on the sample size of the dataset, which is choices13k ($N = 14{,}568$ gambles). **b**–**d**, Comparison between feature values of pHa (**b**), Lb (**c**) and feedback (**d**) with their respective SHAP values. These plots show the structure in the SHAP values for three concrete features, where larger feature values lead to a larger influence on $NN_{difference}$. Every gamble from choices13k is represented as one data point.

by these linear regressions can be seen in Fig. 4a. Overall, basic gamble-describing features could be regressed using SHAP values ($R^2$ between 0.08 and 0.88), while less of the variance in naive and psychological features was linearly explained by the SHAP values ($R^2$ between 0.22 and 0.67 and between 0.14 and 0.54, respectively). This suggests that, while the naive and psychological features capture properties of gambles that are predictive of the differences in choices between the NN trained on CPC15 and choices13k, it is not straightforward to extract these automatically using the SHAP values.

**Theory-driven modelling of the cause of dataset bias**

Taken together, the above analyses establish that the two datasets CPC15 and choices13k do not only differ in their size, that is, the number of gambles they include, but also in the respective behaviour of the participating subjects. The identified features of gambles that could individually best predict the discrepancy in choice behaviour between datasets relate to whether one gamble dominates the respective alternative, or how different the expected values of the gambles are. In all cases, the choice data in choices13k was less extreme, that is, the proportion of choosing gamble A over gamble B was closer to 0.5. One possible reason for such behaviour may be that the decisions by subjects recruited through AMT, and thus contributing to the choices13k dataset, were more variable compared with the subjects contributing to the CPC15 dataset, leading to extreme choice proportions becoming less likely when averaging over all participants. To investigate whether systematic decision noise in the choices13k dataset could account for the discrepancies with the CPC15 dataset, we devised an additional model, 'NN$_{Bourgin,Prior}$ + decision noise', which used the

predictions obtained from $NN_{CPC15}$ and added decision noise. If the hypothesis is correct, that increased decision noise in the choices13k dataset is at least partially responsible for the dataset bias relative to the CPC15 dataset; predictions of this model should better match the data in choices13k and improve performance.

While there is in principle a wide array of possibilities in modelling decision noise[45,46], here we modelled variability of decisions as stemming from two sources: one group of participants guessing randomly and the remaining participants choosing gambles with higher uncertainty or, put differently, according to a process corrupted by additional decision noise (see 'Hybrid probabilistic generative noise model' section in Methods). This model effectively constitutes a hybrid mixture model, in which a proportion $p_{guess}$ of subjects choose one of the two alternative gambles randomly with probability 0.5 while the remaining subjects choose according to the predictions of the NN trained on the CPC15 dataset with additional decision noise. This decision noise was implemented as a multiplicative factor smaller than 1 in log-odds space. The reason for this factor is that this re-scaling of the log odds can be understood as decision noise, or equivalently as a limitation in internal computational precision of the decision[45–47]. How this multiplication transforms probabilities is concretely displayed in Extended Data Fig. 6. The final response rate expressed in this mixture model is a weighted sum of the re-scaled $NN_{CPC15}$ prediction and the random responses by the proportion of subjects who are guessing. We implemented this generative model of human decisions in a hybrid between a NN and Bayesian network and calculated the posterior distribution over the two parameters, that is, the log-odds multiplication factor $f$ and weighting factor $p_{guess}$, using probabilistic programming.

The graphical model is illustrated in Extended Data Fig. 7. More details on the noise model and posterior inference, that is, the fitting of the decision noise, are given in Methods.

The inference of the two latent parameters resulted in both posterior distributions being unimodal and symmetrical, so that we subsequently used the posterior means $p_{guess} = 0.2757$ and $f = 0.6236$ in further analyses. Posterior s.d. were 0.0015 for $p_{guess}$ and 0.0038 for $f$. The full posterior distributions can be seen in Extended Data Fig. 8. Using this model, we again carried out transfer testing by computing the MSE on all datasets. Indeed, this modification in predicting choice proportions leads to an increase in performance on both the choices13k training and testing datasets. Adding decision noise to the NN trained on CPC15 improves the MSE × 100 score from 2.69 to 1.49, that is, by more than 1 and thus closes more than half of the gap between the NNs trained on CPC15 and choices13k. Put differently, modelling 27.6% of the AMT workers in the choices13k dataset as guessing randomly and 72.4% as deciding as the participants from the CPC15 study but with additional log-odds decision noise improves predictions on choices13k.

Among all models trained on CPC15, the hybrid model with decision noise performs best on choices13k. On the other hand, using this modification decreased the scores on the CPC15 dataset by a value of around 1 MSE × 100. Detailed numbers are given in Table 1. Since we mainly compared the predictions of the two NNs, we also checked how similar the $NN_{CPC15}$ with decision noise got to $NN_{choices13k}$. Unsurprisingly, the addition of decision noise led to a decrease in difference between the two models. The MSE × 100 between $NN_{CPC15}$ with decision noise and $NN_{choices13k}$ equals 0.87, or an $R^2$ value of 0.77. In comparison to the linear regressions from Table 2, this level of similarity is even higher than the linear regression using the basic, naive, psychological and HOSD features, despite using only two parameters, the multiplication factor in log-odds space and the mixing weight and optimizing for a proxy.

## Discussion

Understanding and predicting human choices has long been a central goal in economics, psychology, cognitive science and neuroscience, with wide ramifications for the understanding of every-day decisions such as whether to invest, buy or skip goods, services and insurances. While normative models have striven to explain human choices from first principles, descriptive models have tried to capture actual human behaviour, which systematically deviates from the predictions of normative models. Following recent scientific successes involving collecting large datasets and modelling these data with deep NN, two recent studies[16,17] used NNs to better predict human choices on the largest dataset of human risky choices ever collected. Peterson et al.[17] reported finding a NN that implemented 'a policy that outperforms all proposals by human theorists' on their dataset choices13k.

### Dataset difference

In the present study, we revisited this dataset, choices13k, and compared the human decision data on pairs of gambles to two previously collected datasets, CPC15 and CPC18. Our reasoning was that ample previous research has revealed the intricate ways in which data and models can interact, particularly when involving black-box models such as NN[30–34]. Therefore, we systematically investigated the transfer between the two datasets using several models, including the NN architecture used in a previous study[16], which is comparable to the most flexible NN used by Peterson et al.[17], as well as the flexible context-dependent NN model proposed by Peterson et al.[17]. This allowed investigating dataset bias, finding features accounting for differences in predictions between NNs trained on the two datasets and deriving a hybrid probabilistic generative model of how the difference in choice behaviour can be modelled and explained.

Specifically, we first established that there are systematic differences between choices13k and the previously collected laboratory datasets CPC15 and CPC18 by transfer testing[38]. The differences in the

predictions between the NNs trained on the datasets could be captured better by psychological features compared with basic features describing the gambles. However, applying the additive feature attribution method SHAP[36], a popular XAI method to quantify the importance of input feature dimensions towards the decision of an algorithm, did not result in features that could be interpreted in a straightforward way. Instead, we established that stochastic dominance, a concept quantifying how much one gamble should be evaluated as being superior to an alternative, which Kahneman and Tversky described as 'perhaps the most obvious principle of rational choice'[48], was a good predictor of the difference in behaviour across the two NNs and accordingly the two datasets. Indeed, the fact that stochastically dominated gambles are particularly instructive about human choice behaviour has long been known[41] and such gambles therefore have been treated differently, not only in the equivalent model proposed by Peterson et al.[17], but also previously for example in the psychological model BEAST[14]. Because decision proportions for stochastically dominated gambles were always closer to equipreference in choices13k compared with CPC15, we hypothesized that decision proportions in choices13k could be modelled as those from CPC15 but being corrupted by additional decision noise. Accordingly, we devised a probabilistic generative hybrid model in which participants either guess randomly or choose a gamble as participants in the CPC15 study did, but with additional decision noise in log-odds space. Indeed, this model, which essentially deteriorates the decisions of the NN trained on CPC15 by modelling 27.6% of subjects as randomly guessing and the remaining subjects as being corrupted by a decision noise factor of 0.623 in log-odds space, was the best performing model on the choices13k dataset among the models trained on CPC15. This suggests that structured randomness in decisions play a central role in determining the difference in human behaviour between the two datasets.

### Closing the gap

The original study[17] evaluated cross-validated performance of models, that is, it trained models on training data from choices13k and evaluated them on held out testing data to account for the complexity of different models and to avoid overfitting. However, this does not necessarily shield from dataset bias. While Peterson et al. employed a testing set, which was collected through Prolific instead of AMT (supplementary information in ref. [17]), the present analyses establish fundamental differences between the data used in training and testing between CPC15 and choices13k. It is also worth noting that, knowing that black-box NN models' behaviour is commonly difficult to interpret, Peterson et al. devised a hybrid heuristic model, which (1) assigns a fixed, learned choice probability in case the gambles show first order stochastic dominance, which is the case for 16% of all gambles in choices13k. The reason for this choice was that Peterson et al.[17] observed large discrepancies in predictions between trained NN and classic theory in stochastically dominated gambles. The model by Peterson et al.[17] then (2) uses a NN, whose parameters are learned and which compute a weighted average of the predictions stemming from (3) two separate models, which each implement general power utility weighting and Kahneman and Tversky-like probability weightings. However, although a mixture of expert models combining two or more individual models always fit the data equally well or better as individual models[49], our analyses suggest that the particular behaviour in gambles with first order stochastic dominance in choices13k is not necessarily general, as it is distinct from the behaviour in CPC15 in that the choice behaviour contains structured decision noise compared with the laboratory study. Taken together, our re-analysis suggests that there are important differences between the dataset collected under controlled laboratory conditions[14] and the large-scale dataset collected in an online experiment involving AMT workers[17]. Recent analyses of the relationship between datasets collected under different conditions provide a mixed picture. One study[37] suggests that AMT workers largely exhibit similar behaviour as

subjects in laboratory tasks, albeit with increased variability compared with laboratory settings, but at the same time stresses the importance of adequate payment, ensuring task understanding and including attention checks. On the other hand, a recent meta-analysis of a large number of studies involving data collected on AMT paints a grim image of the validity of such datasets[50]. Even more puzzling, it is unclear whether laboratory experiments are the gold standard: attention check tasks have shown that subjects in laboratory experiments also often fail to follow instructions[51], while AMT workers scored better on a similar attention check task compared with subjects in laboratory experiments[52]. Thus, choices13k may allow for the first time large-scale comparison of sophisticated models for human decision-making and ultimately even data-driven model discovery. However, it also opens new challenging questions such as whether the results of Peterson et al.[17] actually transfer to datasets obtained in laboratory experiments, how to explain the wide range of violations of stochastic dominance[41], both in terms of participants[53] and tasks[54], which contextual variables need to be included into decision-making models given the slight differences in protocols across experiments (for example, feedback blocks were repeated only once in ref. [17]), and whether their more theoretically constrained models exhibit better generalization properties than the unconstrained context-dependent NN.

## Conclusion

Although the large-scale dataset presented by Peterson et al. is an impressive achievement and the carefully designed succession of constrained NNs offers enormous potential for uncovering new cognitive and behavioural phenomena, it is difficult to interpret the study as applying deep NNs to a large dataset and thereby automatically discovering a general machine-learned theory of human risky choices that transfers across datasets. The present analyses clearly show that the choice behaviour in choices13k contains structured decision noise compared with the CPC15 laboratory study. And, accordingly, the policy outperforming all proposals by human theorists, incorporates this decision noise. In line with much previous research, our re-analyses and comparisons between datasets show non-trivial relationships between theory, models and data. Thus, our re-analysis adds to the current discussion about the scientific interpretation of NNs' performance exceeding other models. Specifically, the scientific contribution of building NN models of decision-making that reproduce specific choices is not that a model actually does produce these choices, as it should come as no surprise that the function approximation capabilities of NN do succeed when trained to match performance. Discovering the present relationships between data, models and theory was achieved by using theoretical knowledge building on a long history of research in psychology and economics, training and testing of different models including NN across different datasets, and including negative results in the discussion, such as the limited interpretability of the SHAP values.

More generally, the study by Peterson et al.[17] and the current analyses raise a number of important questions for future research, including how decision-making experiments are conducted, what data are collected and which factors influencing people's decisions need to be modelled. Because the datasets considered here only contain the proportion of participants choosing one option in a binary decision, modelling is necessarily limited. Thus, a richer dataset containing individuals' decisions and the order of decisions could allow quantifying individual differences or sequential effects in decisions. Similarly, because the behavioural context, for example whether the experiment was conducted in a laboratory setting or not, seems to play an important role, the question arises how to select, control and quantify different experimental contexts. One exemplary aspect is that the three datasets considered here involving binary risky choices over very small monetary outcomes, with the choice between one certain option and one gamble, account for a very small subset of all risky economic

choices people encounter in their lives. The analyses presented in this study therefore question whether the results of the considered models transfer to decisions involving more gambles[55], more options[56] and larger monetary outcomes[57]. Indeed, several fields including neuroscience and cognitive science have recently advocated more strongly to study decision-making in more naturalistic and embodied contexts[58–60], allowing for inverse modelling of richer datasets[61–64]. On a broader view, these results align with recent theoretical work arguing that theorizing in cognitive science is still not easily automated in an efficient way[65,66]. Similarly, a case study in neuroscience[67] fuels skepticism about possible interpretations that deep learning models discover previously unknown theoretical insights simply by the mere fact of being deep NNs that minimize fitting error[68,69].

## Methods
### Data collection
The three datasets considered in this study were collected in similar experimental paradigms requiring participants to select one of two alternative gambles. All experiments involved binary choice under risk, potential ambiguity and from experience tasks. Subjects were provided with descriptions of two monetary prospects and need to decide between the two.

### CPC15
CPC15 was collected in three separate but methodologically identical experiments, using between 125 and 161 students from the Technion and the HUJI. In each experiment, subjects were faced with 30 gambles, each containing five blocks of five trials each, resulting in 750 decisions per participant. Subjects were paid the earnings of one randomly selected trial plus a show-up fee. This payoff ranged between 10 and 144 shekels (mean of 45.2 shekels). The original paper does not state which board or committee approved their study protocol and whether subjects gave informed consent. More details, especially on the differences between the three experiments, are given in the original paper[14].

### CPC18
CPC18 was again collected using three separate but methodologically identical experiments. Each experiment used 240 subjects, half of which came from Technion and the other half from HUJI. All participants gave informed consent at the beginning of the experiment, and the experimental protocol was approved by Social and Behavioral Sciences Institutional Review Board at the Technion and the Ethics Committee for Human Studies at the Faculty of Agriculture, Food and Environment at HUJI. Again, subjects were paid a show-up fee and the earning of one randomly selected trial. This payment ranged between 10 and 136 shekels (mean of 40 shekels). Additional details, especially on the difference between the three experiments, are given in the original paper[35].

### choices13k
choices13k, on the other hand, was collected on AMT and subjects gave informed consent. The study had institutional review board approval. Subjects were exclusively recruited from the United States and had to have 500 tasks completed with a 95% acceptance rate on the platform. Additionally, participants that had over 80% of their selections as left/right gamble were filtered out, resulting in 14,711 participants. Participants faced 20 gambles, each containing two blocks of five trials. Participants were paid US$0.75 plus a bonus proportional to the reward from a randomly chosen trial. The interface and structure was designed to match CPC's as closely as possible. Additional details are given in the supplementary materials of the original paper[17].

### Datasets
The choices13k as well as the CPC15 datasets used in this paper were used unaltered from their publication. Since the format of the

gambles slightly changed between CPC15, choices13k and CPC18, we reduced CPC18 to the subset of gambles that match the previous format[35]. This selects approximately 90% of the training data and 72% of the test data. It is important to mention that the training data of CPC18 fully contains CPC15's gambles and behavioural data and that additional data for new gambles was collected in the same experimental setup as in CPC15 and also at Technion and the HUJI. The synthetic dataset, synth15, which was used to pre-train the NNs, was sampled using the algorithm described in ref. 14 to match the CPC15 dataset, as described in ref. 16. The psychological features were constructed using the open source code from Plonsky et al.[15]. Our analysis does not exclude any type of gambles (for example, Peterson et al. excluded ambiguous and non-feedback trials[17]).

## Models

In the following, we will discuss details of all models implemented in this study, most of which are re-implementations of models from prior research. In these cases, we highlight if any assumptions were made that could possibly lead to slight differences between the original implementation and our reimplementation. In general, the computational task defined in the CPC[14] that we also adopted here takes the base features describing the gambles as input and predicts the average rate of choosing gamble B over five trials and over all subjects. The SVM and the random forest model additionally take the naive and psychological features as input, as Plonsky et al. have shown that this improves their performance, while the Bourgin et al. NN and BEAST use only base features. Only the NN architecture proposed by Peterson et al.[17] uses a transformation of the input space that consists of explicit pairs of probability and return value of all possible outcomes.

## Bourgin et al. NNs

We use a NN that directly predicts the rate of choosing gamble B, without any additional constraints. This kind of NN is very much comparable with the most expressive class of models used by Peterson et al.[17]. Moreover, the general architecture and training method for the NN models we used was proposed in a study[16] to which all authors of Peterson et al.[17] had contributed. The authors described the architecture of the best performing NN as follows[16]:

> The best multilayer perceptron [...] had three layers with 200, 275, and 100 units, respectively, SReLU activation functions[70], layer-wise dropout rates of 0.15, and an RMSProp optimizer with a 0.001 learning rate. The output layer was one-dimensional with a sigmoid activation function to match the range of the human targets.

Additionally, Bourgin et al. also made use of a training procedure called sparse evolutionary training (SET)[71]. In SET, the layers of the multilayer perceptron start with random sparse connections. After each epoch of training, a fraction of the smallest positive and the largest negative weights are removed. The same number of removed weights gets replaced with new random connections, such that the total number of connections stays constant.

While Bourgin et al. provided detailed information about the architecture, training algorithm and optimizer, some details necessary for reimplementation had to be investigated. We will discuss the details of our training procedure to highlight possible differences due to our reimplementation.

The first deviation occurred at the initialization procedure of the SET algorithm, where Bourgin et al. used a different level of sparsity for every layer, while we used the same for each one, because experiments of adding these additional parameters led to no noticeable improvement.

In cases when we used pre-training, the NN was trained on the synthetic data synth15 generated from BEAST as targets. We found a higher learning rate of $10^{-3}$ to be suitable for the pre-training phase. We optimized the two parameters of the SET algorithm, as well as the batch size using a tree-structured Parzen estimator[72] from the python package hyperopt[73]. For each set of parameters, we trained five different networks from random initialization to reduce the effect of the random training procedure. During pre-training, we trained each network for 300 epochs and saved the weights after every epoch. Later, only the weights with the best validation loss were used for fine-tuning.

During the fine-tuning phase, we used the same batch size as during pre-training and the learning rate of $1 \times 10^{-6}$ that was provided by the authors of the original study. The number of epochs used for fine-tuning were given as 100 for choices13k. For CPC15 we needed more epochs, since the dataset itself is much smaller and hence much fewer updates were made per epoch. The results reported here are based on using 3000 epochs of fine-tuning for CPC15.

## Peterson et al. NNs

We additionally trained the most unconstrained class of NN models from Peterson et al.[17], called context-dependent models. The authors describe the network as follows:[17]

> Our most flexible class of models are neural networks that directly output $P(A)$ given all information about both gambles as input, in our case using two 32-unit hidden layers. Specifically, we define a neural network $g$ such that

$$P(A) = g(x_A, p_A, x_B, p_B),$$

where $x_A$ and $x_B$ are the sets of all possible outcome values and $p_A$ and $p_B$ are the respective probabilities of these outcomes. As previously mentioned, the input features of this NN are different from the base features used by BEAST and the Bourgin et al. NN. The base features are transformed to pairs of probability and value of all outcomes. This transformation therefore does not incorporate the information contained in the base features ambiguity, correlation, feedback and block, which is why Peterson et al. excluded such gambles[17]. However, for a fair comparison among all models we include these gambles in our evaluation and comparisons of all models.

We also used the Adam optimizer with a learning rate of $1 \times 10^{-3}$, as described in the original work. We assumed all units to have sigmoid activation functions because most other models described by Peterson et al.[17] do. Models were trained for 100 epochs on the same 80/20 train–test split as all other models. Optional pre-training on synth15 was done for 20 epochs, using the same learning rate.

Note, that different from the NN model from Bourgin et al.[16], these NNs do not use any form of regularization such as dropout layers or the SET training procedure. Accordingly, we were not able to successfully train this class of models on the CPC15 dataset without overfitting.

## BEAST

The BEAST model is a baseline model for CPC15 introduced by Erev et al.[14]. BEAST predicts gambles by sampling 4,000 agents that are parametrized by six properties, each drawn from uniform distributions. The agents estimate the utility of both gambles by sampling their outcomes, using four different sampling tools. The parameters include the s.d. of the noise term, the number of samples drawn and the tendency for the different samplers. The upper bounds of the uniform distributions were fitted by grid search on the training set of CPC15 to maximize performance. The lower bound for all properties is 0. The choice of a single agent is the sum of the following three factors:

(1) The difference of the best estimates of the expected value of the two gambles: $\text{BEV}_A(r) - \text{BEV}_B(r)$

(2) The difference of the average over $\kappa_i$ samples drawn from the prospect distributions using one of four simulation tools: $ST_A(r) - ST_B(r)$

(3) Gaussian noise $e(r)$ with mean 0 and s.d. $\sigma_i$.

To summarize, an agent decides for gamble A if and only if:

$$[BEV_A(r) - BEV_B(r)] + [ST_A(r) - ST_B(r)] + e(r) > 0$$

Analogously, if the term above is less than 0, the agent decides for gamble B. A property of BEAST that relates to the results of this study concerning stochastic dominance is that the Gaussian noise on the decision of each agent is reduced if one gamble stochastically dominates the other one.

BEAST's sampling tools are designed to reflect four different tendencies. First, the equal weighting tendency towards the option with the best payoff, assuming that all outcomes are equally likely. Second, the tendency to prefer the option that maximizes the probability of the best payoff sign. Third, pessimism as a tendency to assume the worst outcome, and fourth, minimization of immediate regret, which favours options with a low probability of immediate regret. Details regarding the four sampling tools and the influence of the parameters on those can be found in the original work by Erev et al.[14].

There exist multiple variants of BEAST with small adaptations and differences. We used the Python implementation of the original baseline model introduced by Erev et al.[14]. This version differs from later versions implementing individual differences or the changes made for the format of the CPC18.

### SVM and random forest
Both the SVM and the random forest model were implemented using the respective scikit-learn[74] versions. The SVM parameters were all kept at their default values in the library, which includes a radial basis function kernel as well as a regularization value of $C = 1$. Additionally, for the SVM, all features were pre-processed by subtracting the mean and re-scaling the data to unit variance. The random forest includes 500 decision trees, each requiring at least five training data points in each leaf node and used a maximum of four features per split. The parametrizations of these models were chosen such that they match the ones proposed in ref. 15. As mentioned in Table 1, both the SVM and the random forest use naive and psychological features, in addition to the basic gamble features. Plonsky et al. have shown that including them increases the performance of these classical machine-learning models[15]. The performance of both models is similar to the ones originally reported by Plonsky et al.[15].

### Hybrid probabilistic generative noise model
Our noise model is an extension of the NN fine-tuned on CPC15 to better fit the data from choices13k. It incorporates the prediction from $NN_{CPC15}$ as well as noise from two different sources. The first one is a multiplication in log-odds space,

$$logodds = \log\left(\frac{p^{CPC}}{1 - p^{CPC}}\right)$$

$$p_1 = \frac{\exp(logodds \times f)}{1 + \exp(logodds \times f)},$$

where $p^{CPC}$ is the prediction of the NN trained on CPC15. If the factor $f$ is positive, but smaller than 1, this transformation leads to a shift of the predicted probabilities towards 0.5. This type of transformation has been used often to model decision noise or equivalently a limitation on computational precision[46,47]. The second part of our mixture is a random noise term, which assumes that some proportion $p_{guess}$ of the participants are simply guessing whether they choose gamble A or B, with $p_2 = 0.5$.

We model the number of subjects in a particular gamble who are guessing as

$$n_{guess} = p_{guess} \times n,$$

where $n$ is the number of participants who played that particular gamble in the choices13k dataset. The two free parameters ($f$ and $p_{guess}$) of this noise model were estimated on the choices13k training dataset using the probabilistic programming language Turing.jl[75]. For details about the probabilistic model, see Extended Data Figs. 7 and 8 for the posterior distribution of the latent variables. For inference about the posterior distribution, we drew 10,000 samples using the No-U-Turn sampler[76].

### Reporting summary
Further information on research design is available in the Nature Portfolio Reporting Summary linked to this article.

## Data availability
All three datasets, CPC15, CPC18 and choices13k, used in this research, were already available before this study. They can be downloaded under the following links: https://github.com/jcpeterson/choices13k for choices13k, https://economics.agri.huji.ac.il/crc2015/raw-data for CPC15 and https://cpc-18.com/data/ for CPC18.

## Code availability
All code, including pre-trained model weights, needed to replicate the results in this study is publicly available in the GitHub repository at https://github.com/RothkopfLab/DatasetBias.

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

## Acknowledgements
We thank the authors of the studies 'Cognitive model priors for predicting human decisions'[16] and 'Using large-scale experiments and machine learning to discover theories of human decision-making'[17], particularly D. Bourgin and J. Peterson, for extensive help in re-implementing their models and fruitful discussion. C.A.R. and K.K. acknowledge the support of the Hessian research priority program LOEWE within the project 'WhiteBox'. F.T., C.A.R. and K.K. are supported by the cluster project 'The Adaptive Mind'. To.T., C.A.R. and K.K.'s research is supported by the cluster project 'The Third Wave of AI' as part of the Excellence Program of the Hessian Ministry of Higher Education, Science, Research and Art. Finally, we thank the anonymous reviewer 3 and F. Jäkel for useful comments to the paper. The funders had no role in study design, data collection and analysis, decision to publish or preparation of the manuscript.

## Author contributions
To.T., D.S., F.T., M.S., Tü.T. and C.A.R. designed the research. To.T., F.T., M.S. and Tü.T. carried out the experiments. To.T., D.S., F.T., M.S., Tü.T., K.K. and C.A.R. analysed the data. To.T., D.S., F.T., C.A.R. and K.K. have written and revised the paper.

## Competing interests
The authors declare no competing interests.

## Additional information
**Extended data** is available for this paper at https://doi.org/10.1038/s41562-023-01784-6.

**Correspondence and requests for materials** should be addressed to Tobias Thomas.

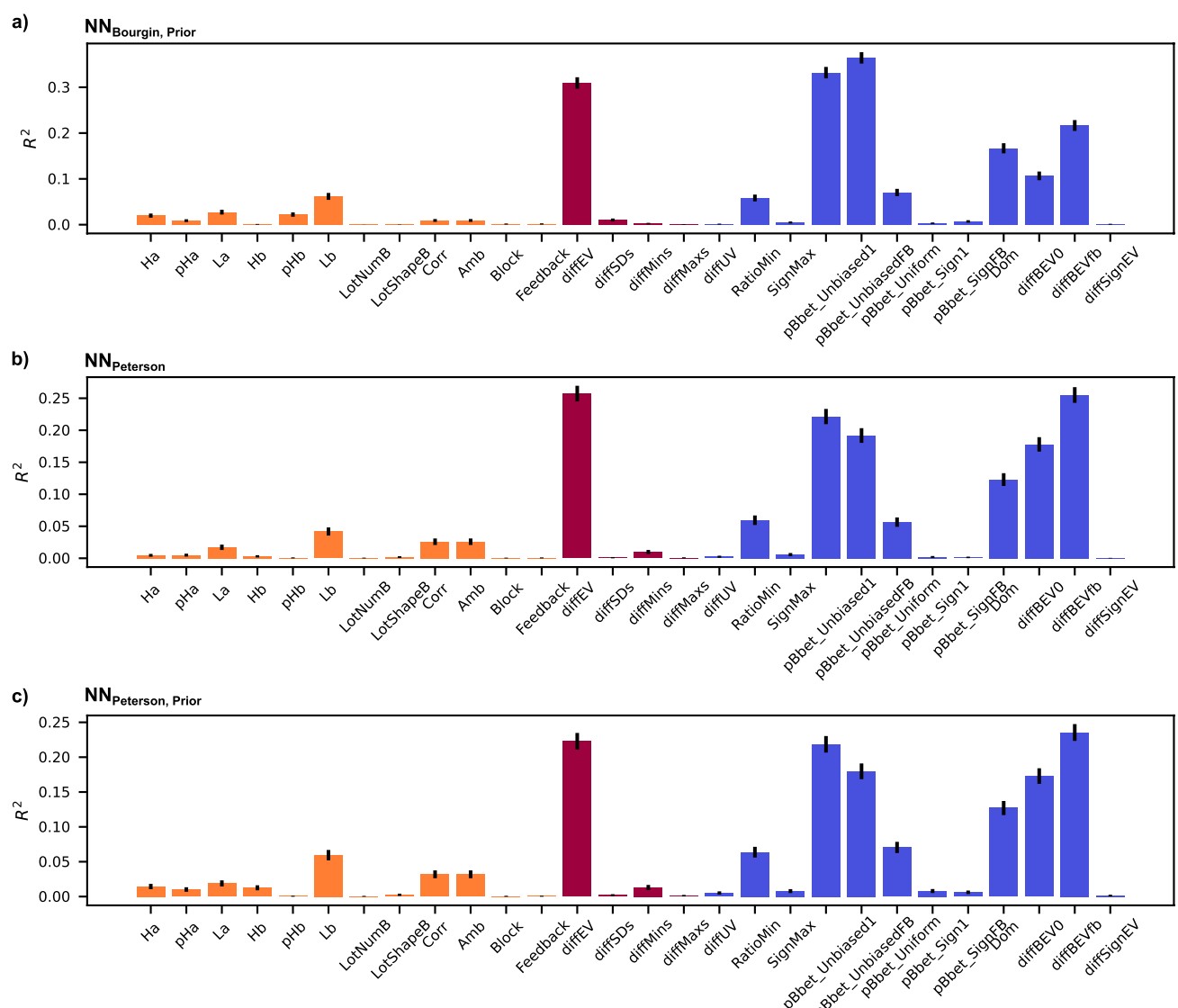

**Extended Data Fig. 1 | Relationship between single features and NN_difference for all networks.** Repeating the analysis from Figure [2a] for $NN_{Bourgin,Prior}$ (a), $NN_{Peterson}$ (b), and $NN_{Peterson,Prior}$ (c) as the NN model trained on choices13k. Errorbars still represent the 95% confidence interval, which are based on the sample size, which is choices13k (N = 14568 gambles).

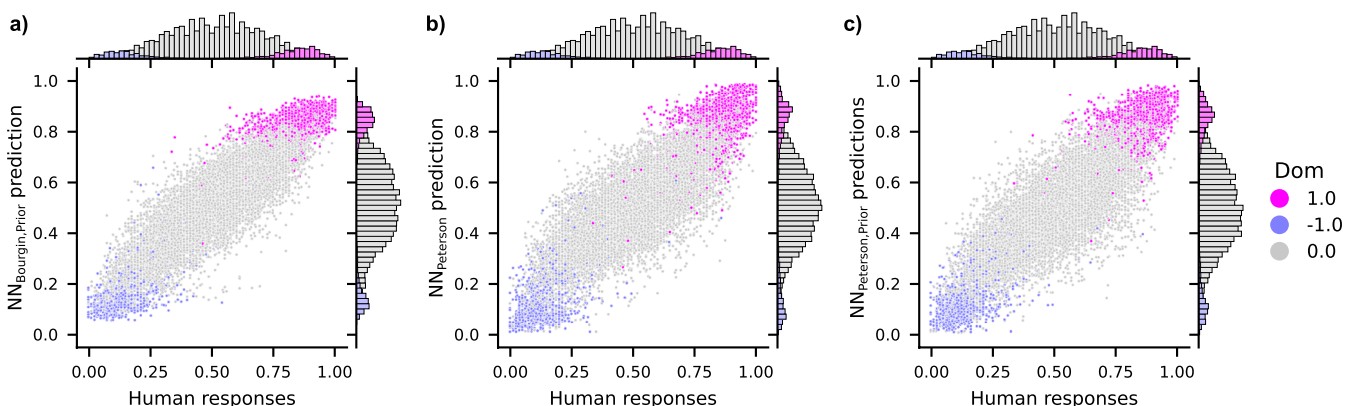

**Extended Data Fig. 2 | Influence of Dominance on human behavior and NN predictions.** Repeating the analysis from Figure 3b for $NN_{Bourgin,Prior}$ (a), $NN_{Peterson}$ (b), and $NN_{Peterson,Prior}$ (c) as the NN model trained on choices13k. Figure 3a,c were not repeated because they do not depend on the model trained on choices13k.

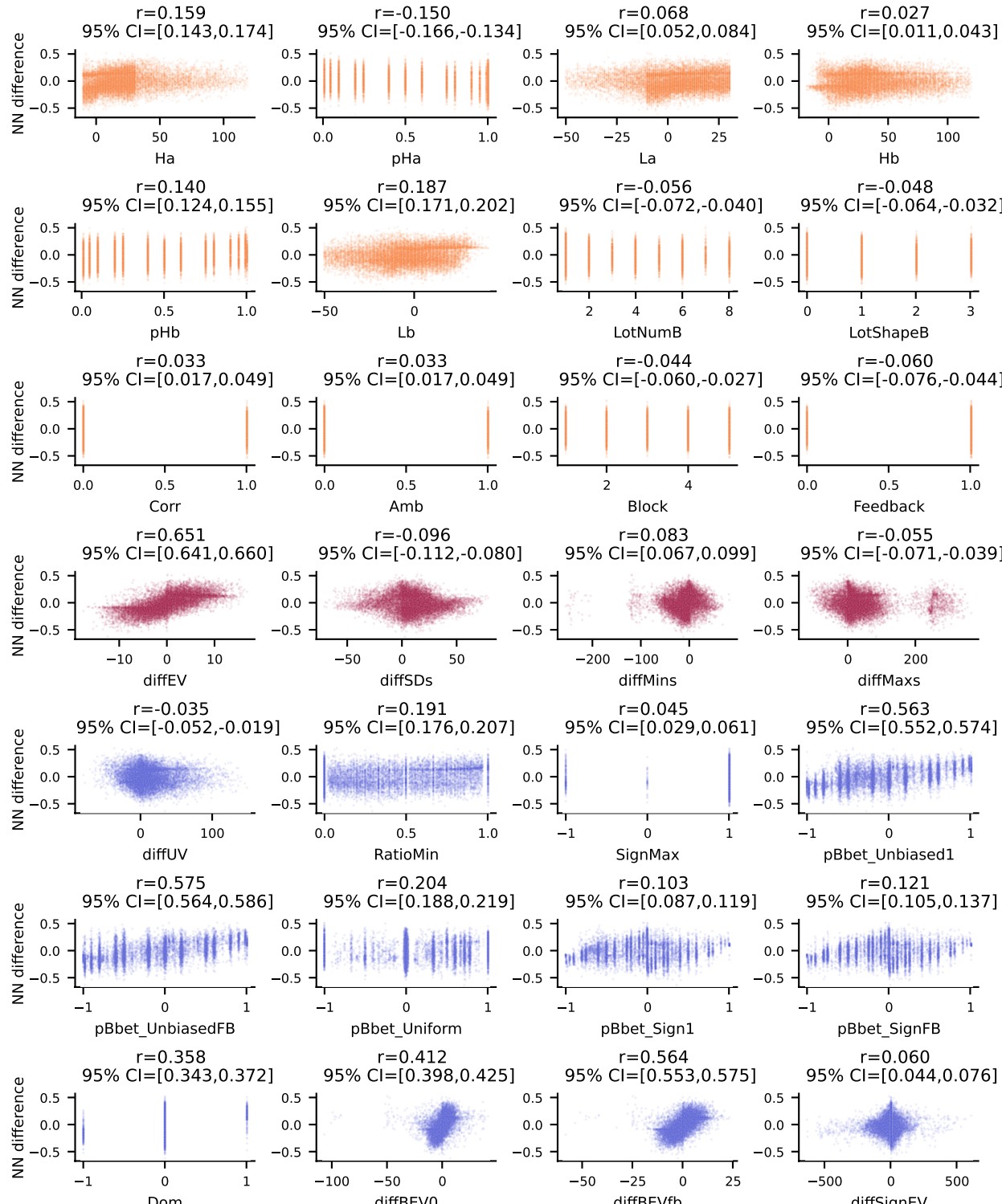

**Extended Data Fig. 3 | Relationship between features and NN$_{difference}$.**
Every small plot visualizes one feature on the x-axis and the difference in NN predictions on the y-axis. Every decision problem corresponds to a single point. The color of the points represent the type of feature (orange = basic, red = naive, blue = psychological, for a detailed explanation see[16]). The title indicates the Pearson correlation coefficient between the two properties as well as the 95% confidence interval of the correlation coefficient.

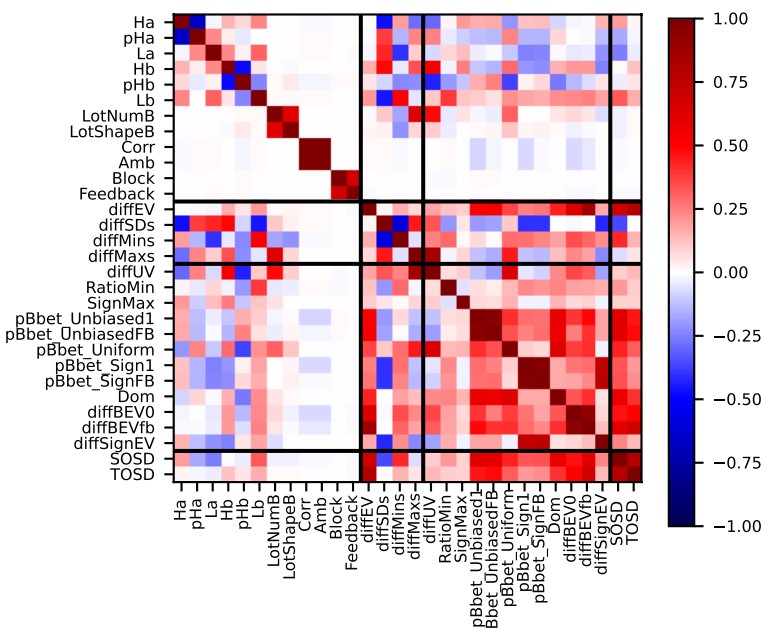

**Extended Data Fig. 4 | Feature correlation matrix.** Pairwise Pearson correlation coefficient matrix between all features. Data to measure this correlation were taken from the choices13k dataset. The black lines separate types of features. From left to right (and top to bottom), basic features, naive features, psychological features and higher orders of stochastic dominance. Note that, beyond the features proposed in previous studies and employed by Plonsky et al.[16], we additionally computed second (SOSD) and third order stochastic dominance (TOSD).

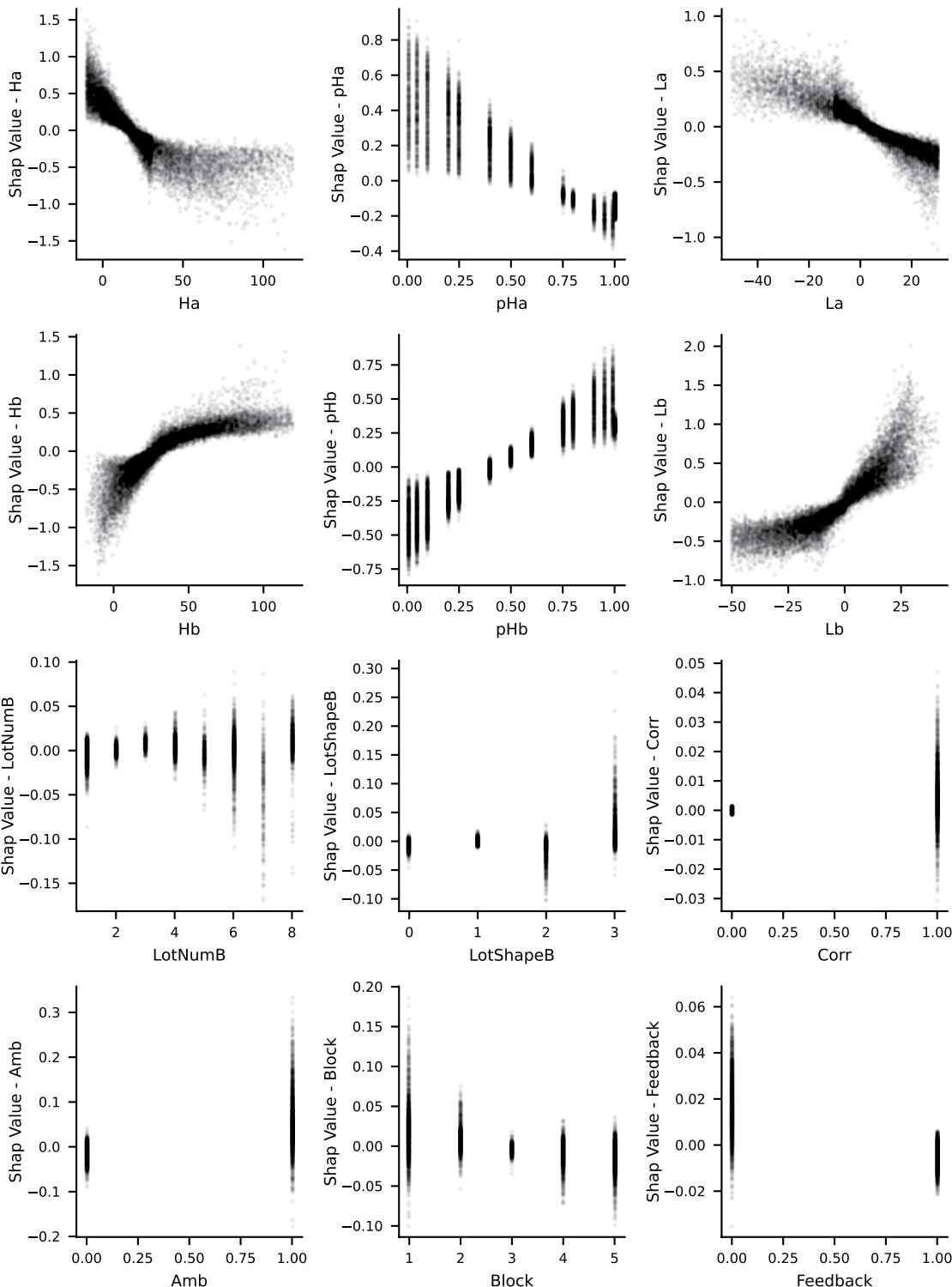

**Extended Data Fig. 5 | Relationship between NN input feature values and their respective SHAP value.** The plots show each choices13k gamble as one dot. The x-axis is one of basic gamble (NN input) features, while the y-axis is the corresponding SHAP value from the difference between both NNs.

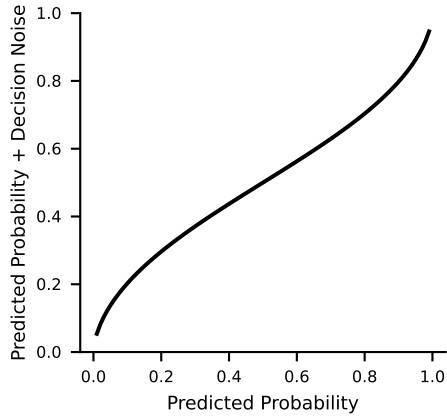

**Extended Data Fig. 6 | Probability transformation.** Displayed is the transformation of a probability by multiplying in log-odd space. The multiplication factor is 0.6236 - the posterior mean of the model shown in Figure S7. The x-axis is the probability, while the y-axis shows the corresponding probability after multiplying it in log-odd space.

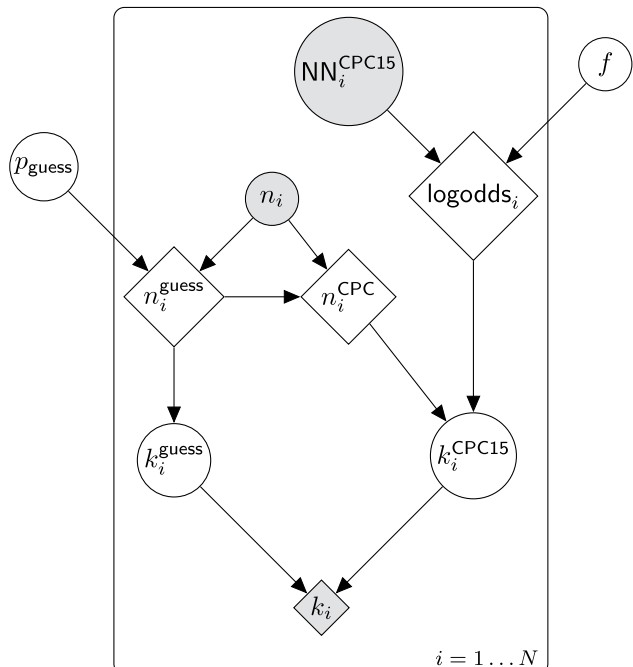

**Extended Data Fig. 7 | Bayesian network of the noise model.** Clear nodes represent latent variables, whereas shaded nodes represent observed quantities. Variables inside diamonds are deterministic given its parents, while the ones inside circles are random variables. The plate $i = 1 \ldots N$ is over the number of gambles in the choices13k dataset. The model has two parameters, the proportion of participants assumed to be guessing randomly in each trial $p_{\text{guess}}$ and the rescaling factor in log-odds space f to account for the decision noise of the remaining participants. The prior for both parameters were chosen to be Beta(1, 1), that is uniform over the [0, 1] interval. The distribution of the sum of two Binomial random variables needed to compute the distribution of ki was approximated using the Gaussian approximation to the Binomial. The predictions $NN_i^{CPC15}$ in this model are from the NN finetuned on CPC15 and the observed responses $k_i$ are the human responses on choices13k.

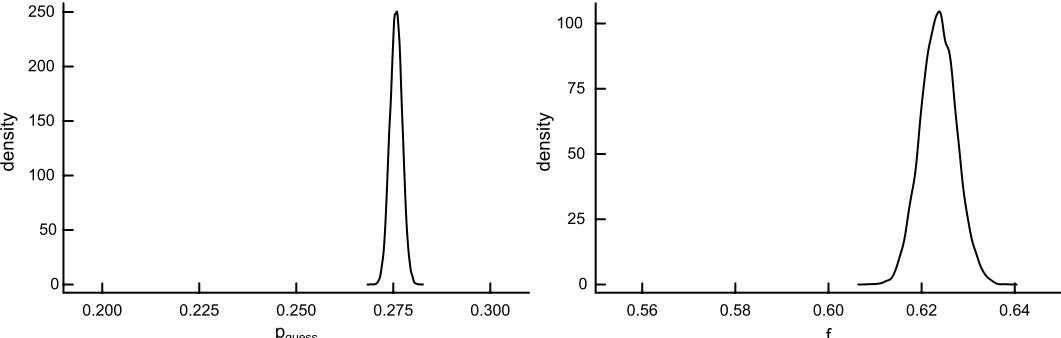

**Extended Data Fig. 8 | Parameter posterior distributions.** Posterior distribution over the two parameters from the model shown in Figure S7.

**Extended Data Table 1 | Table 2 for other NNs trained on choices13k**

| Features | Base | | Base+Naive | | Base+Naive+Psych. | | Base+Naive+Psych.+HOSD | |
|---|---|---|---|---|---|---|---|---|
| NN Name | MSE | $R^2$ | MSE | $R^2$ | MSE | $R^2$ | MSE | $R^2$ |
| $NN_{Bourgin, Prior}$ | 0.0163 | 0.1376 | 0.0110 | 0.4206 | 0.0081 | 0.5716 | 0.0080 | 0.5764 |
| $NN_{Peterson}$ | 0.0203 | 0.0982 | 0.0151 | 0.3275 | 0.0123 | 0.4540 | 0.0121 | 0.4608 |
| $NN_{Peterson, Prior}$ | 0.0170 | 0.1291 | 0.0134 | 0.3125 | 0.0109 | 0.4422 | 0.0107 | 0.4496 |

Linear regressions between different sets of features and $NN_{difference}$ for different NN models trained on choices13k. This is a repetition of the analysis from Table 2 three more NN models trained on choices13k.

**Extended Data Table 2 | Description of all gamble features**

| Feature type | Name | Description |
|---|---|---|
| Basic | Ha | High payoff value of gamble A |
| | pHa | Probability for the high payoff option of gamble A |
| | La | Low payoff value of gamble A |
| | Hb | High payoff value of gamble B |
| | pHb | Probability for the payoff option of gamble B |
| | Lb | Low payoff value of gamble B |
| | LotNumB | Number of options of the low payoff lottery of gamble B |
| | LotShapeB | The shape of the low payoff lottery of gamble B |
| | Amb | Indicates whether pHb is observable for the participants |
| | Corr | The correlation between the outcomes of the two gambles |
| | Feedback | Did subjects receive feedback about the outcome of the non chosen option |
| Naive | diffEV | Difference between expected value of the gambles |
| | diffSDs | Difference between the standard deviations of the gambles |
| | diffMins | Difference between the minimal outcomes of the gambles |
| | diffMaxs | Difference between the maximal outcomes of the gambles |
| Psychological | diffBEV0 | Estimator for the EV difference if B is ambiguous |
| | diffBEVfb | Estimator for the EV difference if B is ambiguous and after receiving feedback |
| | pBbet_Unbiased1 | Probability of B generating a higher outcome |
| | pBbet_UnbiasedFB | Probability of B generating a higher outcome, after receiving feedback |
| | diffUV | Difference between EV under the assumption that all outcomes are equally likely |
| | pBbet_Uniform | Probability for B being better under the assumption that all outcomes are equally likely |
| | diffSignEV | Difference between expected values using only the sign of each outcome |
| | pBbet_Sign1 | Probability that diffSignEV $> 0$ in ambiguous cases |
| | pBbet_SignFB | Probability that diffSignEV $> 0$ in ambgious cases and after receiving feedback |
| | SignMax | Signals whether there is a possibility for a positive outcome |
| | RatioMin | Ratio between the minimal value of the gambles if their sign is equal, 0 otherwise |
| | Dom | Indicates, whether one gamble stochastically dominates the other gamble |

The basic features are necessary to describe a gamble, while all the other ones are transformations of basic features. For further details on all features and the respective background literature on the psychological ones, see[16].

# Reporting Summary

## Statistics

For all statistical analyses, confirm that the following items are present in the figure legend, table legend, main text, or Methods section.

| n/a | Confirmed | |
|---|---|---|
| ☐ | ☒ | The exact sample size (*n*) for each experimental group/condition, given as a discrete number and unit of measurement |
| ☐ | ☒ | A statement on whether measurements were taken from distinct samples or whether the same sample was measured repeatedly |
| ☒ | ☐ | The statistical test(s) used AND whether they are one- or two-sided<br>*Only common tests should be described solely by name; describe more complex techniques in the Methods section.* |
| ☒ | ☐ | A description of all covariates tested |
| ☐ | ☒ | A description of any assumptions or corrections, such as tests of normality and adjustment for multiple comparisons |
| ☐ | ☒ | A full description of the statistical parameters including central tendency (e.g. means) or other basic estimates (e.g. regression coefficient) AND variation (e.g. standard deviation) or associated estimates of uncertainty (e.g. confidence intervals) |
| ☒ | ☐ | For null hypothesis testing, the test statistic (e.g. *F*, *t*, *r*) with confidence intervals, effect sizes, degrees of freedom and *P* value noted<br>*Give P values as exact values whenever suitable.* |
| ☐ | ☒ | For Bayesian analysis, information on the choice of priors and Markov chain Monte Carlo settings |
| ☒ | ☐ | For hierarchical and complex designs, identification of the appropriate level for tests and full reporting of outcomes |
| ☐ | ☒ | Estimates of effect sizes (e.g. Cohen's *d*, Pearson's *r*), indicating how they were calculated |

*Our web collection on statistics for biologists contains articles on many of the points above.*

## Software and code

Policy information about availability of computer code

| Data collection | We have not collected any data, so no software was used. |
|---|---|
| Data analysis | python 3.8.10, numpy 1.18.5, pandas 1.3.5, scipy 1.4.1, tensorflow 2.3.0, scikit-learn 1.0.2, shap 0.37.0, hyperopt 0.2.3, Julia 1.7.0, Turing.jl 0.21.13 |

For manuscripts utilizing custom algorithms or software that are central to the research but not yet described in published literature, software must be made available to editors and reviewers. We strongly encourage code deposition in a community repository (e.g. GitHub). See the Nature Portfolio guidelines for submitting code & software for further information.

## Data

Policy information about availability of data

All manuscripts must include a data availability statement. This statement should provide the following information, where applicable:
- Accession codes, unique identifiers, or web links for publicly available datasets
- A description of any restrictions on data availability
- For clinical datasets or third party data, please ensure that the statement adheres to our policy

All three datasets, CPC15, CPC18, as well as choices13k, used in this research, were already available prior to this study.
They can be downloaded under the following links: https://github.com/jcpeterson/choices13k for choices13k, https://economics.agri.huji.ac.il/crc2015/raw-data for CPC15, and https://cpc-18.com/data/ for CPC18.

## Research involving human participants, their data, or biological material

Policy information about studies with human participants or human data. See also policy information about sex, gender (identity/presentation), and sexual orientation and race, ethnicity and racism.

| | |
|---|---|
| Reporting on sex and gender | Not applicable |
| Reporting on race, ethnicity, or other socially relevant groupings | Not applicable |
| Population characteristics | Not applicable |
| Recruitment | Not applicable |
| Ethics oversight | Not applicable |

Note that full information on the approval of the study protocol must also be provided in the manuscript.

# Field-specific reporting

Please select the one below that is the best fit for your research. If you are not sure, read the appropriate sections before making your selection.

☐ Life sciences    ☒ Behavioural & social sciences    ☐ Ecological, evolutionary & environmental sciences

For a reference copy of the document with all sections, see nature.com/documents/nr-reporting-summary-flat.pdf

# Behavioural & social sciences study design

All studies must disclose on these points even when the disclosure is negative.

| | |
|---|---|
| Study description | All three datasets used in this study (CPC15, CPC18 and choices13k) were previously collected and already publicly available. All three share the same experimental paradigm of economic binary choice under risk, potential ambiguity and from experience. Subjects were faced with descriptions of two monetary prospects and needed to decide between the two. Both CPC15 and CPC18 were recorded in three seperate experiments, using between 125 and 240 students from Technion and HUJI. choices13k on the other hand, was recorded online on Amazon Mechanical Turk, using 14711 participants from all over the US. More details are given in the paper as well as in the original papers, introducing these datasets. |
| Research sample | Not applicable |
| Sampling strategy | Not applicable |
| Data collection | Not applicable |
| Timing | Not applicable |
| Data exclusions | Not applicable |
| Non-participation | Not applicable |
| Randomization | Not applicable |

# Reporting for specific materials, systems and methods

We require information from authors about some types of materials, experimental systems and methods used in many studies. Here, indicate whether each material, system or method listed is relevant to your study. If you are not sure if a list item applies to your research, read the appropriate section before selecting a response.

## Materials & experimental systems

| n/a | Involved in the study |
|-----|----------------------|
| ☒ | Antibodies |
| ☒ | Eukaryotic cell lines |
| ☒ | Palaeontology and archaeology |
| ☒ | Animals and other organisms |
| ☒ | Clinical data |
| ☒ | Dual use research of concern |
| ☒ | Plants |

## Methods

| n/a | Involved in the study |
|-----|----------------------|
| ☒ | ChIP-seq |
| ☒ | Flow cytometry |
| ☒ | MRI-based neuroimaging |

