## [Peer Review File · Nature Human Behaviour]

Peer Review Information

Journal: Nature Human Behaviour

Manuscript Title: Modeling data set bias in machine-learned theories of economic decision making

Corresponding author name(s): Tobias Thomas

Reviewer Comments & Decisions:

Decision Letter, initial version:
--

30th March 2023

Dear Mr Thomas,

Thank you once again for your manuscript, entitled "Still no free lunch from deep learning: Data, models, and theory of human economic decision-making", and for your patience during the peer review process. Please accept my sincere apologies for the delay in the peer review of your work.

Your Article has now been evaluated by 2 referees. We also solicited signed comments by the authors of the original research your work critiques, which are provided in an attached PDF.

Please note that signed comments do not function as reviews. The authors of signed comments are asked to comment solely on methodological aspects of your work, as well as on the extent to which their work is accurately represented in your manuscript. Decisions on whether to reject a manuscript or invite a revision are based on the feedback of our independent reviewers. However, if a decision to invite a revision is made, signed comments inform our requests for necessary revisions. (For full details on our Signed Comments policy, please see <https://www.nature.com/nathumbehav/signed-comments>.)

You will see from the comments of our 2 independent reviewers copied below that, although they find your work of potential interest, they have raised quite substantial concerns. The signed comments also raise some important issues. In light of these comments, we cannot accept the manuscript for publication, but would be interested in considering a revised version if you are willing and able to fully address reviewer and editorial concerns, in particular the methodological issues raised by Reviewer #1.

We hope you will find the referees' comments useful as you decide how to proceed. If you wish to submit a substantially revised manuscript, please bear in mind that we will be reluctant to approach the referees again in the absence of major revisions. We are committed to providing a fair and constructive peer-review process. Do not hesitate to contact us if there are specific requests from the reviewers that you believe are technically impossible or unlikely to yield a meaningful outcome.

If you wish to submit a suitably revised manuscript we would hope to receive it within 4 months. I would be grateful if you could contact us as soon as possible if you foresee difficulties with meeting this target resubmission date.

- Include a "Response to the editors and reviewers" document detailing, point-by-point, how you addressed each editor and referee comment. If no action was taken to address a point, you must provide a compelling argument. When formatting this document, please respond to each reviewer comment individually, including the full text of the reviewer comment verbatim followed by your response to the individual point. This response will be used by the editors to evaluate your revision and sent back to the reviewers along with the revised manuscript.
- You should also include responses to the signed comments.
- Highlight all changes made to your manuscript or provide us with a version that tracks changes.

[REDACTED]

Thank you for the opportunity to review your work. Please do not hesitate to contact me if you have any questions or would like to discuss the required revisions further.

Sincerely,
Jamie

Dr Jamie Horder
Senior Editor
Nature Human Behaviour

REVIEWER COMMENTS:

Reviewer #1:

Remarks to the Author:

Two recent papers (Bourgin et al., 2019 and Peterson et al, 2021) suggest that certain neural net models provide more accurate prediction of human decision making than models that explicitly build on psychological theory. The current paper extends these analyses and shows an important shortcoming of the neural net model it examines: While this model provides best predictions within each of the studied data sets, it provides less accurate generalization between data sets (relative to a random forest model with features designed based on psychological theory). In addition, the paper clarifies the existence of clear differences between the data sets it examines that contribute to the failure of the neural net model.

I think that the question addressed by the current paper is highly important, and believe that the main hypotheses are interesting and reasonable. Yet, I also think that the analysis suffers from important shortcomings.

Assuming that I correctly understand the analysis, the most important shortcoming involves the nature of the neural net model examined here (based on Bourgin et al., 2019). I think that the estimation of this model based on CPC15 used the psychological model (BEAST) to derive syntactic training data (or “cognitive model prior”). Thus, it is not a theory-free model. It is natural to assume (as implied by Bourgin et al., and by the CPC15 competition) that the effort to capture CPC15 with a pure theory-free model would lead to less accurate predictions than the model that relies on BEAST priors. This issue should be clarified.

I think that Bourgin et al. estimated their neural net model based on the choice13k data in two ways: with and without the BEAST priors. Their results suggest that the BEAST priors (that implies reliance on psychological theory) help, but the magnitude of the help decreases with the number of observations. Only when the model is estimated on 80% of the data (more than 10,000 games), the BEAST priors do not appear to improve prediction accuracy. The current analysis appears to focus on the estimation that uses

the BEAST priors. To allow a clear test of the authors' hypothesis, they should also test Bourgin et al. model when it does not use the BEAST priors.

To facilitate a fair test of the models with and without the BEAST priors, the authors should focus on the test set of CPC15 (as the BEAST priors and the random forest models were derived based on the training set of CPC15). To add data, I suggest that they will repeat the analyses for the CPC18 data.

Other suggestions:

The authors should clarify the fact that their analysis focuses on Bourgin et al.'s model, and do not consider the model proposed in Peterson et al. Peterson et al. focus on a subset of the choice13 data (no ambiguity, with feedback)

The authors write: "Note that all these features directly relate to how much better one option is on average relative to the other." I think that this statement is confusing, as these features capture the probability that one option is better than the other (and not the expected value difference).

I think that the statement (line 151) "which is comparable to the most expressive model presented by Peterson et al." is incorrect. Please check.

Reviewer #2:

Remarks to the Author:

The study reanalyses two large datasets designed to validate and develop further cognitive science theories related to human decisionmaking. The authors importantly point to interrelationships of theory and hypothesis-driven data generation and their analysis, and conclude that fully automated development of theories is not to be expected from Neural Networks. I have minor comments:

- The manuscript would benefit from a few more subheadings as the relevant information e.g. on specifics of data generation of the original datasets is a bit spread across the manuscript. The point of how human participants generated the data under specific conditions, i.e. AMT, could be reinforced, particularly as theoretical rationale for introducing the modeling of decision noise.

- Specifically the theoretical assumptions on decisionmaking behavior that were modelled, citing from the manuscript: "here we modeled decision noise as stemming from two sources: one group of participants guessing randomly and the remaining participants potentially choosing gambles with additional decision noise", could be extended to help the reader understand better what "potentially choosing gambles with additional decision noise" means.

SIGNED COMMENTS:

Signed Comments regarding Thomas et al.

Thomas et al., in response to our work (Bourgin et al., 2019; Peterson et al., 2021), put forward a “central claim that theory generation still cannot be outsourced entirely to powerful machine-learning black-boxes” and “that careful combination of theory and data analysis is required to understand human risky choices”. We do not disagree with either of these claims. We believe both of our papers are consistent with these ideas, and certainly don’t present them as claims to be challenged. Bourgin et al. (2019) demonstrates that *theory is useful* even to those interested purely in prediction because insights from theories can be transferred to machine learning models via learning from their predictions. Peterson et al. (2021) illustrates that carefully designed limitations to neural network model architectures can be used to ask *theoretical questions* about datasets and evaluate entire classes of theories as opposed to testing one theory at a time, helping us to develop new psychological theories based on the resulting insights. The end output of Peterson et al. (2021) is a Mixture of Theories model (MOT) that integrates past theoretical disagreements in the risky choice literature. The argument presented by Thomas et al. is thus dependent on a reading of these papers that is not characteristic of our views.

Thomas et al. conduct an analysis that sets out to compare our dataset (choices13k) to a previous one (cpc15). (As we discuss below, this analysis also incorporates a very large synthetic dataset, synth15, generated from an existing psychological model of decision-making, which makes some of these comparisons harder to interpret.) Because each dataset contains different choice problems, they cannot be compared directly, so Thomas et al. instead compare the predictions of machine learning models trained on each dataset. They find the difference between the two sets of predictions can be partially explained by (a) noise, and (b) predictors developed by Plonsky et al. (2017) based on psychological theory.

We first address (a). Thomas et al. observe that the noise in our data (at the level of choice proportions per decision problem) is higher than cpc15. This is to be expected, as we obtained choices from fewer participants per problem than cpc15 (i.e., ~15 vs. >100), as reported in our work. We chose to focus on scaling the number of problems instead of precision within single problems because larger sets of problems provide more opportunities to falsify and contrast theories. Plonsky et al. (2021) have already made a similar observation about noise levels and modeled this noise to improve the predictions of their own theory (BEAST). While this can

improve performance on some datasets, Plonsky et al. (2021) also point out that it is not enough: their theory (which was developed to explain cpc15) results in predictions that are “too rational” and thus worse than our model.

We now address (b). BEAST, which was featurized into the “Psychological Features” employed in the current work by its creators (Plonsky et al.), was originally developed to explain cpc15. Peterson et al. (2021) evaluated BEAST and found it outperformed previous theories, which makes sense given that it is a member of the class of theories we identified as having the necessary level of complexity to explain decision behavior. Thus, our original results have similar implications and are analogous to what Thomas et al. find: that knowledge based on cpc15, BEAST, and “Psychological Features” is relevant to making generally good predictions. But again, just as Thomas et al. find 18% of the variance in the differences between a cpc15- and choices13k-trained network cannot be explained, we found that the gap between the two is not fully closed. In Peterson et al. (2021), we aimed to close this gap by more precisely linking theory to resulting prediction improvements (via the novel MOT theory), while Thomas et al. focus on linear models representing feature importance.

Thomas et al. also evaluate transfer performance (performance of a network trained on one dataset on the other dataset) and find that, when looking only at raw MSE scores, some models appear overfit to their respective datasets. However, differences in MSE across datasets are less meaningful than the transfer of core theoretical principles, which was our transfer metric of choice. We were also concerned about transfer performance, and to address this Peterson et al. (2021) presented a second new dataset obtained using a different method (with more problems per participant) and population from choices13k. Like cpc15, this dataset is also meaningfully different from choices13k ($r=0.80$), yet when models trained on choices13k were tested on this new one, our central results were replicated: the ordering of models was preserved even though there was an increase in MSE due to transfer. That is, while behavior may drift between datasets and task designs, we obtained the same answers to theoretical questions about behavior: problem-level context effects are central (Context-Dependent Theories), gamble-level context effects are second most central (Value-Based Theories), and likewise for all other major conclusions.

Beyond the considerations above, there are also technical issues with the analysis comparing our dataset to cpc15. The claim that training on the choices13k dataset produces similar results to training on cpc15 with noise relies on pretraining on synth15, inspired by our work in Bourgin et al. (2019). However, as illustrated in Bourgin et al. (2019), pretraining on a theory means the network knows the insights from that theory and thus does not need to learn it from data. Bourgin et al. (2019) explicitly illustrated that theory and data are interchangeable in this way.

This implies that the cpc15-trained network presented by Thomas et al. is biased to be similar to one trained on choices13k: the former learned about behavior from a leading theory, and the latter learned it from data alone. This may create the illusion to the reader that cpc15 has the same informativeness as choices13k because both networks have taken different routes to learn similar things. In Peterson et al. (2021) our focus was on networks trained exclusively on human data, which is where the larger dataset is beneficial, and we believe that without pretraining on synth15 the cpc15-trained network would behave quite differently from that trained on choices13k. The analysis presented by Thomas et al. is unable to explain 18% of the variance in the differences between a CPC- and choices13k-trained network, but given the dependence on synth15 this number is likely to be significantly underestimated.

As a result of their analysis, Thomas et al. claim that choices13k is biased, implying that others are not. We take the view that all datasets are biased. The main impetus for our work was to avoid what we view as the worst kind of bias: that introduced by only looking at a small portion of human behavior. Our large dataset aimed at minimizing this kind of bias and encouraging theories that can explain decisions for as many choice problems as possible. Another way to reduce a different kind of bias is to collect a dataset like cpc15 which focuses more on increasing the precision of exact choice proportions within a problem. As discussed above, we also collected a second dataset with the aim of increasing precision at the level of participants. However, the other possible sources of bias or differences between datasets are myriad. One strength of Peterson et al. (2021) is that it proposed a method that can be reapplied to produce new theories for datasets collected from different contexts, populations, and task structures.

We appreciate the efforts of Thomas et al. to understand the relationships between different datasets and models, and we think such efforts are important. However, we believe that there are significant issues with the way they frame this analysis as a response to claims we wouldn't endorse, the specific comparisons they present between models maximizing the apparent similarity between those models, and the possible interpretation of their conclusions about noise, transfer performance, and bias being a result of a lack of care in our own analyses rather than intrinsic consequences of the statistical structure of the task that we worked hard to mitigate.

**Joshua Peterson,
Mayank Agrawal,
David Bourgin,
Daniel Reichman,
& Thomas Griffiths**

References

Plonsky, O., & Erev, I. (2021). To predict human choice, consider the context. *Trends in Cognitive Sciences*, 25(10), 819-820.

Plonsky, O., Erev, I., Hazan, T., & Tennenholtz, M. (2017). Psychological forest: Predicting human behavior. In *The Proceedings of the Thirty-first AAAI Conference on Artificial Intelligence*.

Author Rebuttal to Initial comments**Reviewer 1:**

Two recent papers (Bourgin et al., 2019 and Peterson et al, 2021) suggest that certain neural net models provide more accurate prediction of human decision making than models that explicitly build on psychological theory. The current paper extends these analyses and shows an important shortcoming of the neural net model it examines: While this model provides best predictions within each of the studied data sets, it provides less accurate generalization between data sets (relative to a random forest model with features designed based on psychological theory). In addition, the paper clarifies the existence of clear differences between the data sets it examines that contribute to the failure of the neural net model.

We would like to thank the reviewer for their positive evaluation of the scope and the results of our manuscript.

I think that the question addressed by the current paper is highly important, and believe that the main hypotheses are interesting and reasonable. Yet, I also think that the analysis suffers from important shortcomings.

Assuming that I correctly understand the analysis, the most important shortcoming involves the nature of the neural net model examined here (based on Bourgin et al., 2019). I think that the estimation of this model based on CPC15 used the psychological model (BEAST) to derive syntactic training data (or “cognitive model prior”). Thus, it is not a theory-free model. It is natural to assume (as implied by Bourgin et al., and by the CPC15 competition) that the effort to capture CPC15 with a pure theory-free model

would lead to less accurate predictions than the model that relies on BEAST priors. This issue should be clarified.

The assumptions made by the reviewer are fully correct. The reason why the NN models we used in the first version of the manuscript rely on pretraining is that they would otherwise overfit the training dataset of CPC15 because of its small size. Thus, they would have even better scores on the training dataset, but their transfer performance to all other datasets would be impaired considerably. This was already observed in the original studies by Bourgin et al. (2019) and Peterson et al. (2021) and we can confirm these findings. We clarify this now in the section 'Choice models and training' in the revised version of the manuscript.

Nevertheless, this does not compromise our main conclusion about generalization between data sets, for several reasons. First, the other machine learning models (SVM, random forest) perform similarly on CPC15 as the NNs, but also do not generalize from CPC15 to choicesk13k, although one might argue that they are also not theory free due to the use of hand-designed features. Second, the psychological model BEAST, which was used to generate the synthetic training dataset synth15, was tweaked specifically to mimic the behavior of subjects in CPC15 already in the original publication. Therefore, pretraining on the synthetic data does not weaken the conclusion that a model trained on CPC15 does not generalize to choices13k. Third, we now also include test transfer in the other direction, from choices13k to CPC15. In this case, pretraining is not necessary, and we include both versions of the models with and without pretraining in the revised version of the manuscript. These results further unequivocally support our conclusions. See also our answer to the next point for further details.

To address this point in the manuscript, we made changes to the text in the results section, specifically subsection 'Choice models and training', 'Establishing dataset bias', and in the Methods section, specifically subsection 'Peterson et al. Neural Networks'.

I think that Bourgin et al. estimated their neural net model based on the choice13k data in two ways: with and without the BEAST priors. Their results suggest that the BEAST priors (that implies reliance on psychological theory) help, but the magnitude of the help decreases with the number of observations. Only when the model is estimated on 80% of the data (more than 10,000 games), the BEAST priors do not appear to improve prediction accuracy. The current analysis appears to focus on the estimation that uses the BEAST priors. To allow a clear test of the authors' hypothesis, they should also test Bourgin et al. model when it does not use the BEAST priors.

Yes, thank you. We fully agree that this additional test helps to clarify any potential influences of the pretraining on synth15, i.e., the "cognitive model prior". Accordingly, to address this point, we

trained additional NNs on the choices13k dataset. First, we did what you, Reviewer 1, suggested and trained the Bourgin et al. (2019) NN on choices13k without pretraining on the BEAST priors. This NN (in the style of Bourgin et al. (2019) without pretraining on BEAST-generated synthetic data synth15) performs significantly worse on the test set of CPC15 and the transfer to other data sets is impaired compared to the neural network with synth15 pretraining, yet the performance on choices13k is comparable. This confirms our previous conclusions.

Note, however, that our analysis on transfer between CPC15 and choices13k holds also for the other machine learning models (SVM, random forest), which do not need pretraining to achieve a similar performance as the neural network (see Table 1). Additionally, the results also hold for the models based on Peterson et al. (2021) with and without BEAST pretraining, which we further describe below in our answer to your question about these models. Thus, both these additional results further support the conclusion that dataset bias is present.

Accordingly, we adjusted the analyses in the current version of the main text and the figures to use this new network based on Bourgin et al. (2019) without prior training on synthetic data, but repeated all analyses and provide the associated figures with the other NNs trained on choices13k in the supplementary material.

All these additional analyses show relatively moderate quantitative differences, but all our results and conclusions regarding dataset bias as quantified by transfer testing were confirmed so that our arguments were supported further.

To address this point, we made changes to the manuscript in the results section, specifically subsection 'Choice models and training', 'Establishing dataset bias', 'Data-driven analysis of dataset bias', Table 1 and in the Methods section, specifically subsection 'Peterson et al. Neural Networks' as well as in the Supplementary Information.

To facilitate a fair test of the models with and without the BEAST priors, the authors should focus on the test set of CPC15 (as the BEAST priors and the random forest models were derived based on the training set of CPC15). To add data, I suggest that they will repeat the analyses for the CPC18 data.

Thank you for this suggestion. The additional analysis on CPC18 is a bit tricky, though. The reason is a slight change in the format of the gambles between CPC15 (and choices13k) and CPC18. In CPC18, the low return of gamble A could also be a lottery. In CPC15 as well as in choices13k, this is only possible for gamble B. To account for this, we extracted the subset of CPC18 gambles that have the same format as CPC15 gambles. This leads to 92% of the training set and 70% of the test set being usable for additional transfer testing. The results are shown in the updated Table 1 in the current version of the manuscript.

These results suggest that human behavior is very similar between CPC15 and CPC18 because models performing well on one of the two datasets also perform well on the respective other one. By contrast, models trained on choices13k still do not generalize well to the CPC15 and also do not generalize well to CPC18, with comparable MSE between these two datasets.

To address this point, we made changes to the manuscript in the results section, specifically subsection 'Establishing dataset bias', Table 1 and in the Methods section, specifically subsection 'Datasets'.

Other suggestions:

The authors should clarify the fact that their analysis focuses on Bourgin et al.'s model, and do not consider the model proposed in Peterson et al. Peterson et al. focus on a subset of the choice13 data (no ambiguity, with feedback)

We fully agree with the reviewer that our analyses in the previous version of the manuscript were limited to the neural network model proposed in Bourgin et al. (2019). The main reason has been that the models proposed in Peterson et al. (2021) have not been published till today. However, in light of the points raised by the reviewer, we have decided to also implement the context-dependent NN of Peterson et al. (2021) to the best of our knowledge, based on the given descriptions in the original paper. As we were able to match the performance measures on choices13k reported in Peterson et al. (2021), we are confident that our NNs are close to the ones in the original study.

Accordingly, we have now clarified that some analyses are based on Bourgin et al.'s (2019) model. Additionally, we have now trained NNs based on Peterson et al. (2021), which are trained on the same subset of choices13k, which was used in the original paper, i.e. no ambiguity and with feedback. While we were able to reproduce the performance values of these models on choices13k reported in Peterson et al. (2021), to provide a fair comparison to all the other models, we also evaluate this model on the full training and test set of all datasets, including gambles with ambiguity and without feedback. Note that these analyses were excluded in Peterson et al. (2021). The respective transfer performances are shown in the updated Table 1 in the manuscript. Importantly, all these additional results further support the conclusion that there is dataset bias between choices13k and CPC15.

To address this point, we made changes to the manuscript in the results section, specifically subsection 'Choice models and training', 'Establishing dataset bias', 'Data-driven analysis of dataset bias', Table 1 and in the Methods section, specifically subsection 'Peterson et al. Neural Networks'.

The authors write: “Note that all these features directly relate to how much better one option is on average relative to the other.” I think that this statement is confusing, as these features capture the probability that one option is better than the other (and not the expected value difference).

Yes, thank you, the sentence was indeed confusingly worded. We have changed it to “the degree to which one gamble is expected to yield a higher payoff than an alternative.”

I think that the statement (line 151) “which is comparable to the most expressive model presented by Peterson et al.” is incorrect. Please check.

The most expressive class of models in Peterson et al. is the so-called context dependent NN class. These NNs are given the properties of both gambles as input and output the probability to choose gamble A. This is also how the NN from Bourgin et al., which we had used exclusively in the previous version of the manuscript, works. The main difference between the two models is that Peterson et al.’s inputs are explicit probability, value pairs of all possible outcomes, while Bourgin et al. use 12 descriptive features, which also implicitly contain these distributions. We clarify this in the description of the respective NNs in the Methods section. We hope that this clarifies this point satisfactorily.

Reviewer 2:

The study reanalyses two large datasets designed to validate and develop further cognitive science theories related to human decisionmaking. The authors importantly point to interrelationships of theory and hypothesis-driven data generation and their analysis, and conclude that fully automated development of theories is not to be expected from Neural Networks. I have minor comments:

We would like to thank the reviewer for their positive evaluation of the scope and the results of our manuscript.

- The manuscript would benefit from a few more subheadings as the relevant information e.g. on specifics of data generation of the original datasets is a bit spread across the manuscript. The point of how human participants generated the data under specific conditions, i.e. AMT, could be reinforced, particularly as theoretical rationale for introducing the modeling of decision noise.

We have tried to address this point of improving the structure of the manuscript by adding several subheadings to the results and discussion sections.

To address the second point, instead of only mentioning the rationale of previous research demonstrating increased variability in AMT data late in the text in the results section, we now additionally directly mention it in the introduction.

- Specifically the theoretical assumptions on decisionmaking behavior that were modelled, citing from the manuscript: "here we modeled decision noise as stemming from two sources: one group of participants guessing randomly and the remaining participants potentially choosing gambles with additional decision noise", could be extended to help the reader understand better what "potentially choosing gambles with additional decision noise" means.

Yes, thank you. This was too compressed in the previous version of the manuscript. We have clarified this sentence in the results section, specifically the subsection 'Theory driven modeling of the cause of data set bias' and now also refer to the relevant additional information in the Methods section, which describes the noise model in detail and explains the psychological meaning of the model's variables.

Signed Comment:

Thomas et al., in response to our work (Bourgin et al., 2019; Peterson et al., 2021), put forward a "central claim that theory generation still cannot be outsourced entirely to powerful machine-learning black-boxes" and "that careful combination of theory and data analysis is required to understand human risky choices". We do not disagree with either of these claims. We believe both of our papers are consistent with these ideas, and certainly don't present them as claims to be challenged. Bourgin et al. (2019) demonstrates that theory is useful even to those interested purely in prediction because insights from theories can be transferred to machine learning models via learning from their predictions. Peterson et al. (2021) illustrates that carefully designed limitations to neural network model architectures can be used to ask theoretical questions about datasets and evaluate entire classes of theories as opposed to testing one theory at a time, helping us to develop new psychological theories based on the resulting insights. The end output of Peterson et al. (2021) is a Mixture of Theories model (MOT) that integrates past theoretical disagreements in the risky choice literature. The argument presented by Thomas et al. is thus dependent on a reading of these papers that is not characteristic of our views.

We are sorry if we raised the impression that our reading of these papers appears not to reflect the views of the authors. However, we would like to point out that we are not alone in interpreting the article by Peterson et al. (2021) as suggesting that theory generation can be outsourced to machine-learning algorithms, as emphasized by the commentary published in Science alongside the original article, which we also cite in our manuscript:

“Peterson et al. (4) demonstrate the power of a more recent approach: Instead of relying on the intuitions and (potentially limited) intellect of human researchers, the task of theory generation can be outsourced to powerful machine-learning algorithms” (Bhatia & He, 2021).

Moreover, this seems to be in line with the abstract by Peterson et al. (2021), which states that “progress toward this goal [of understanding decision-making] can be accelerated by using large datasets to power machine-learning algorithms that are constrained to produce interpretable psychological theories.”

Except for these different possible interpretations of the work by Peterson et al. (2021), we are not in disagreement with anything stated here. We agree that neural networks can be used to ask questions about datasets. Indeed, this is precisely what we do in the present manuscript. Combining data, models, and theory leads us to conclude that the “Mixture of Theories model” is based on modeling a dataset, which systematically and significantly deviates from previous decision datasets obtained in laboratory settings.

Thomas et al. conduct an analysis that sets out to compare our dataset (choices13k) to a previous one (cpc15). (As we discuss below, this analysis also incorporates a very large synthetic dataset, synth15, generated from an existing psychological model of decision-making, which makes some of these comparisons harder to interpret.) Because each dataset contains different choice problems, they cannot be compared directly, so Thomas et al. instead compare the predictions of machine learning models trained on each dataset. They find the difference between the two sets of predictions can be partially explained by (a) noise, and (b) predictors developed by Plonsky et al. (2017) based on psychological theory.

We first address (a). Thomas et al. observe that the noise in our data (at the level of choice proportions per decision problem) is higher than cpc15. This is to be expected, as we obtained choices from fewer participants per problem than cpc15 (i.e., ~15 vs. >100), as reported in our work. We chose to focus on scaling the number of problems instead of precision within single problems because larger sets of problems provide more opportunities to falsify and contrast theories. Plonsky et al. (2021) have already made a similar observation about noise levels and modeled this noise to improve the predictions of their own theory (BEAST). While this can improve performance on some datasets, Plonsky et al. (2021) also point out that it is not enough: their theory (which was

developed to explain cpc15) results in predictions that are “too rational” and thus worse than our model.

We agree that differences between datasets are an important factor, as we show, and we also agree that this has already been acknowledged by previous research. However, we do not simply add “noise” to the data but instead devise a hybrid probabilistic generative model that provides a structured and interpretable noise model, making two different sources of noise in the dataset explicit. Using this model and the data, we estimated the parameters of the two sources of noise through probabilistic programming. This analysis therefore does not simply add noise but models noise in two distinct ways: the probability that a proportion of participants answered randomly and the amount of psychologically motivated decision noise in log-odds space for those participants answering in a similar way as the participants in the CPC15 dataset. Both of these factors are readily interpretable and have psychological meaning. Importantly, this model explicitly accounts for the number of participants that provided choice data separately for every choice problem using the Binomial distribution.

In our manuscript, we additionally reference psychological as well as economic literature that has found widespread support for the approach of a multiplication in log-odds space as a sound method of modeling noise in human decisions. Importantly, the analysis involving the probabilistic generative model of the fraction of participants guessing and the amount of decision variability in log-odds space can now be incorporated in future analyses of such datasets and we believe this to be a valuable scientific contribution.

We now address (b). BEAST, which was featurized into the “Psychological Features” employed in the current work by its creators (Plonsky et al.), was originally developed to explain cpc15. Peterson et al. (2021) evaluated BEAST and found it outperformed previous theories, which makes sense given that it is a member of the class of theories we identified as having the necessary level of complexity to explain decision behavior. Thus, our original results have similar implications and are analogous to what Thomas et al. find: that knowledge based on cpc15, BEAST, and “Psychological Features” is relevant to making generally good predictions. But again, just as Thomas et al. find 18% of the variance in the differences between a cpc15- and choices13k-trained network cannot be explained, we found that the gap between the two is not fully closed. In Peterson et al. (2021), we aimed to close this gap by more precisely linking theory to resulting prediction improvements (via the novel MOT theory), while Thomas et al. focus on linear models representing feature importance.

The analysis reported in the subsection ‘Data-driven analysis of dataset bias’ involving the psychological features, which are also used in BEAST, is primarily intended to show that such

an analysis is of limited use in explaining the reason for the dataset bias between CPC15 and choices13k. We did not use these features to make “generally good predictions”, as the correlations between individual psychological features and the difference in predictions between different NNs is rather low. It is also not the case that we find that “the difference between the two sets of predictions can be partially explained by (...) predictors developed by Plonsky et al. (2017) based on psychological theory”.

Instead, the results of the analyses involving the different types of features motivated the analyses in the subsequent section “Theory-driven identification of dataset bias”. Thus, we do not “focus on linear models representing feature importance” but instead use the analyses involving the linear features to go back to psychological and economic theory, specifically human behavior in gambles with first, second, and third order stochastic dominance. This motivated us to derive a hybrid probabilistic generative model of choices that involves structured variability and is able to relate the differences in choices between the CPC15 dataset, obtained at the Hebrew University of Jerusalem and the Technion, and the choices obtained through Amazon Mechanical Turk.

Thomas et al. also evaluate transfer performance (performance of a network trained on one dataset on the other dataset) and find that, when looking only at raw MSE scores, some models appear overfit to their respective datasets. However, differences in MSE across datasets are less meaningful than the transfer of core theoretical principles, which was our transfer metric of choice.

We are sorry, but “the transfer of core theoretical principles” needs to be quantified somehow. Peterson et al. (2021) themselves used MSE to evaluate and quantify performance, and it is the de facto standard in the literature of predicting human economic decision-making, including the CPC15 and CPC18 competitions. The analysis of the mixture components of the MOT model may provide additional theoretical insights, but devising the MOT is based on selecting the context-dependent NN based on the MSE across the different models in Peterson et al. (2021). Thus, to quantify how well theoretical principles, such as generalization between datasets, apply to different datasets, we also used the MSE.

We were also concerned about transfer performance, and to address this Peterson et al. (2021) presented a second new dataset obtained using a different method (with more problems per participant) and population from choices13k. Like cpc15, this dataset is also meaningfully different from choices13k ($r=0.80$), yet when models trained on choices13k were tested on this new one, our central results were replicated: the ordering of models was preserved even though there was an increase in MSE due to transfer. That is, while behavior may drift between datasets and task designs, we obtained the

same answers to theoretical questions about behavior: problem-level context effects are central (Context-Dependent Theories), gamble-level context effects are second most central (Value-Based Theories), and likewise for all other major conclusions.

First of all, it is reassuring that the authors acknowledge that they are also concerned about transfer performance, which is what we systematically evaluated in our current manuscript. We would have liked to also evaluate our models on this additional dataset, which the authors describe in Peterson et al. (2021) as having been collected on Prolific, but this dataset seems to be unavailable to the public.

We acknowledge that it is an important result that the order of respective models in terms of the MSE stayed the same on these two datasets, as stated by the authors. However, it is much less clear what this means scientifically, particularly if the ordering follows the models' expressivity: it should not come as a surprise that context-dependent theories, which the authors correctly described as "our most general class of functions" in the original publication, fit data best, when allowing individual separate data fits. It is expected to be the best fitting model on all datasets. Thus, while the ordering may be maintained, it is currently not clear whether the mixture of theories, MOT, derived from the best fitting model would result in the same mixture across all considered datasets. Given the present results, this is quite unlikely.

Beyond the considerations above, there are also technical issues with the analysis comparing our dataset to cpc15. The claim that training on the choices13k dataset produces similar results to training on cpc15 with noise relies on pretraining on synth15, inspired by our work in Bourgin et al. (2019). However, as illustrated in Bourgin et al. (2019), pretraining on a theory means the network knows the insights from that theory and thus does not need to learn it from data. Bourgin et al. (2019) explicitly illustrated that theory and data are interchangeable in this way. This implies that the cpc15-trained network presented by Thomas et al. is biased to be similar to one trained on choices13k: the former learned about behavior from a leading theory, and the latter learned it from data alone.

Please accept our apologies but we must disagree here. The empirical results provide unequivocal evidence that there is (!) a difference between the models trained on the two datasets and that they are indeed not (!) similar. The claim that "theory and data are interchangeable" does not hold empirically. Otherwise the networks trained on choices13k would transfer to the CPC15 and CPC18 datasets. Thus, all models trained on CPC15 show bias when tested on choices13k and all models trained on choices13k show bias when tested on CPC15, demonstrating dataset bias. And this bias is reduced, when adding structured noise

both from assuming a proportion of participants guessing randomly and the remaining participants deciding with added decision noise.

This may create the illusion to the reader that cpc15 has the same informativeness as choices13k because both networks have taken different routes to learn similar things. In Peterson et al. (2021) our focus was on networks trained exclusively on human data, which is where the larger dataset is beneficial, and we believe that without pretraining on synth15 the cpc15-trained network would behave quite differently from that trained on choices13k. The analysis presented by Thomas et al. is unable to explain 18% of the variance in the differences between a CPC- and choices13k-trained network, but given the dependence on synth15 this number is likely to be significantly underestimated.

There are a number of statements here that need to be addressed individually.

First, we did not pretrain on a theory but on data, the synth15 dataset. This dataset was generated by a model, BEAST, which was derived from psychological theory, but whose parameters were optimized (grid-searching approximately 20,000 hyperparameter settings to find the best multilayer perceptron) so as to generate data as closely as possible to the empirical choice data in cpc15 (see Erev et al. (2017)). Thus, pretraining on synth15 may bias a NN to produce choices that tend to be similar to those in CPC15 and not necessarily the entire theory: this is an empirical question that requires proper testing, as we did.

Second, the influence of pretraining a NN on the synth15 dataset needs to be quantified. This is what we did already in the previous version of the manuscript. In the current version, we expanded these analyses to show that the assertions made by the authors here do not hold under empirical testing. To estimate the effect of pretraining, we now trained both the NN from Bourgin et al. (2019) and the context-dependent NN from Peterson et al. (2021), each once with and once without pretraining. The results are shown in Table 1. Both networks show very similar performance on the train and the test set of choices13k, while the NNs with pretraining generalize slightly better to CPC15: the improvements are from 1.25 to 0.9 and from 1.31 to 1.27. These results show that pretraining models before fine-tuning on choices13k indeed only has a small effect.

All the other analyses done in the paper were also repeated for the different NN models trained on choices13k and are shown in the supplementary materials. The results change only quantitatively, yet qualitatively all results and arguments stay the same, regardless whether the choices13k NN was pretrained or not, or which architecture it uses.

Third, for models that are pretrained on synth15 and subsequently fine-tuned on CPC15, on the other hand, this pretraining is very important to avoid overfitting. This was already acknowledged by Bourgin et al. (2019) and we can confirm this observation for all the NNs we tested. However, the synth15 dataset was generated to emulate the choices in CPC15. Thus, if “theory and data” were “interchangeable” as stated by the authors, transfer testing should show no dataset bias when transfertesting. Instead, we find clear dataset bias, including for the models that do not use pretraining on synth15, namely SVM and random forest.

As a result of their analysis, Thomas et al. claim that choices13k is biased, implying that others are not. We take the view that all datasets are biased. The main impetus for our work was to avoid what we view as the worst kind of bias: that introduced by only looking at a small portion of human behavior. Our large dataset aimed at minimizing this kind of bias and encouraging theories that can explain decisions for as many choice problems as possible. Another way to reduce a different kind of bias is to collect a dataset like cpc15 which focuses more on increasing the precision of exact choice proportions within a problem. As discussed above, we also collected a second dataset with the aim of increasing precision at the level of participants. However, the other possible sources of bias or differences between datasets are myriad. One strength of Peterson et al. (2021) is that it proposed a method that can be reapplied to produce new theories for datasets collected from different contexts, populations, and task structures.

It was not our intention to claim that choices13k is more biased than any other dataset, and we are sorry if that impression arose due to our writing. We fully agree that in a Bayesian framework, all datasets are expected to be biased. We also agree with many of the other statements here. However, we maintain that, scientifically, it is important to tease apart and quantify at least some of the “myriad” “differences between datasets” in order to understand human decision-making, particularly if conclusions are based on fitting models to one dataset with very specific biases. And one strength of our analyses is that they can be reapplied to different datasets from different contexts, populations, and tasks.

We appreciate the efforts of Thomas et al. to understand the relationships between different datasets and models, and we think such efforts are important. However, we believe that there are significant issues with the way they frame this analysis as a response to claims we wouldn't endorse, the specific comparisons they present between models maximizing the apparent similarity between those models, and the possible interpretation of their conclusions about noise, transfer performance, and bias being a result of a lack of care in our own analyses rather than intrinsic consequences of the statistical structure of the task that we worked hard to mitigate.

We thank the authors for the positive and constructive points they raised and hope that we were able to answer all points satisfactorily. We did certainly not imply at any point in our manuscript that the authors of the original studies Bourgin et al. (2019) and Peterson et al. (2021) demonstrated “lack of care” regarding analyses. We carefully went through the manuscript and made small but specific changes both in the introduction and in the discussion to avoid the impression of insinuating “lack of care” by the authors. In our experience, every scientific result not only answers some questions, but additionally raises some new questions. We added a short passage in the discussion emphasizing that because of the Peterson et al. (2021) study, new scientific questions can now be addressed quantitatively. In our analyses, we addressed some of the questions regarding the nature of the impressive dataset, choices13k, which the authors have collected and analyzed.

Decision Letter, first revision:

13th September 2023

Dear Dr. Thomas,

Thank you for your patience as we’ve prepared the guidelines for final submission of your Nature Human Behaviour manuscript, "Still no free lunch from deep learning: Data, models, and theory of human economic decision-making" (NATHUMBEHAV-22102718A). Please carefully follow the step-by-step instructions provided in the attached file, and add a response in each row of the table to indicate the changes that you have made. Please also address the additional marked-up edits we have proposed within the reporting summary. Ensuring that each point is addressed will help to ensure that your revised manuscript can be swiftly handed over to our production team.

We would hope to receive your revised paper, with all of the requested files and forms within two-three weeks. Please get in contact with us if you anticipate delays.

If you have not done so already, please alert us to any related manuscripts from your group that are under consideration or in press at other journals, or are being written up for submission to other

journals (see:

<https://www.nature.com/nature-research/editorial-policies/plagiarism#policy-on-duplicate-publication> for details).

Nature Human Behaviour offers a Transparent Peer Review option for new original research manuscripts submitted after December 1st, 2019. As part of this initiative, we encourage our authors to support increased transparency into the peer review process by agreeing to have the reviewer comments, author rebuttal letters, and editorial decision letters published as a Supplementary item. When you submit your final files please clearly state in your cover letter whether or not you would like to participate in this initiative. Please note that failure to state your preference will result in delays in accepting your manuscript for publication.

In recognition of the time and expertise our reviewers provide to Nature Human Behaviour's editorial process, we would like to formally acknowledge their contribution to the external peer review of your manuscript entitled "Still no free lunch from deep learning: Data, models, and theory of human economic decision-making". For those reviewers who give their assent, we will be publishing their names alongside the published article.

Cover suggestions

We welcome submissions of artwork for consideration for our cover. For more information, please see our https://www.nature.com/documents/Nature_covers_author_guide.pdf target="new"> guide for cover artwork.

ORCID

Non-corresponding authors do not have to link their ORCIDs but are encouraged to do so. Please note that it will not be possible to add/modify ORCIDs at proof. Thus, please let your co-authors know that if they wish to have their ORCID added to the paper they must follow the procedure described in the following link prior to acceptance:

Nature Human Behaviour has now transitioned to a unified Rights Collection system which will allow our Author Services team to quickly and easily collect the rights and permissions required to publish your work. Approximately 10 days after your paper is formally accepted, you will receive an email in providing you with a link to complete the grant of rights. If your paper is eligible for Open Access, our Author Services team will also be in touch regarding any additional information that may be required to arrange payment for your article.

Please note that *Nature Human Behaviour* is a Transformative Journal (TJ). Authors may publish their research with us through the traditional subscription access route or make their paper immediately open access through payment of an article-processing charge (APC). Authors will not be required to make a final decision about access to their article until it has been accepted. Find out more about Transformative Journals

Please use the following link for uploading these materials:
[REDACTED]

Best regards,
Alex McKay
Editorial Assistant
Nature Human Behaviour

On behalf of

Jamie

Dr Jamie Horder
Senior Editor
Nature Human Behaviour

Reviewer #1:

Remarks to the Author:

The revised draft addresses my main concerns with the original manuscript. I think that the paper highlights an interesting observation. The existence of large differences between the decisions people make in different experiments (the data set bias), suggests that running large experiments does not guarantee good generalization of the results to different settings. To facilitate generalization (and useful predictions) it is important to clarify the difference between the different data sets. The paper shows that it is not easy to achieve this goal with traditional ML methods, and demonstrates a natural theoretic analysis can help.

The paper could be improved by adding analyses that examine the value of the theoretical idea it proposes. In theory, it is possible that the explanatory value of the added noise hypothesis is only a good post-hoc explanation of the current findings. Yet, I think that this analysis can wait to future research.

A minor comment: Figure 1 should be updated to include the CPC18 analyses

Reviewer #3:

Remarks to the Author:

Review of "Still no free lunch from deep learning: Data, models, and theory of human economic decision-making" NHB

This paper reanalyzes several aspects of prominently published work by Peterson et al (2021, Science) building on a very similar 2019 Bourgin paper.

I agree with the previous Commenters who thought that some of the language in this paper-- including the "no free lunch" in the title, which the authors must get rid of-- is rejecting a claim that has rarely been made.

Instead, the main contribution of this paper is to dig into how much dataset bias there might be in Peterson et al, and what causes it. This is a very interesting question because "dataset bias" is a well-known challenge in learning from datasets but is not often recognized by that name in experimental

social science.

This comment is therefore useful in extending scientists' understanding of the predictive scope of this paper and how to explore its limits, empirically.

The dataset bias referred to is whether an NN or other theoretical extrapolation from one dataset (say choice13k in this case) generalizes well in predicting an unseen different dataset.

The value of machine learning-enhanced theory discovery presumably depends, as do all other kinds of theory generation, on dataset bias not being too large and unpredictable. (If dataset bias is large, that is manageable as long as there is some theory or guidelines about when to expect poor generalization.) So the question of dataset bias is absolutely central to the general point Peterson et al were making about the usefulness of the large data set approach. However, it is also clear that dataset bias clearly was not the focus of Peterson et al's paper at all (there term is not even mentioned in their paper).

However, concern for dataset bias is important in thinking about what ideas derived from the analysis of specific data sets would generalize to very different kinds of data sets used to test risky choice modeling. Many of the Thomas et al authors and the original Peterson et al authors may not be aware that in the larger space of possible experimental (and natural) choices involving financial risk, the dataset differences between cpc15, cpc18, and choice13k are small. They all use binary choices between two money gambles with small amounts of money, and not too many outcomes, which are presented as descriptions. Subjects choose one choice in a pair and can't express a strength of preference or indifference. Even within the large domain of financial risk experiments, and risky choices in everyday life, there is hardly any daylight between the cpc sets and choice13k.

So if there is dataset bias here, as measured by the low transferability of predictions derived from one of these datasets to another, that is surely a lower bound on how bad the data set bias might be for very different tasks, amounts of money, and very likely subject populations. I will discuss this further below at location *SCOPE*

You should think about a retitling such as "An investigation into dataset bias in machine-learned theory discovery"

I am seeing this ms. for the first time along with the authors' rebuttal letter that includes original referee report questions they tried to answer. My comments are therefore in two groups.

The first batch is specific comments about improving the paper, which is mostly editorial.

The second batch of comments is my attempt to 'mediate' between the "signed commenter" comments and the authors' responses.

First comments batch about the Thomas et al paper.

Some citations are a bit odd. For instance, citation one cites Glimcher on deviations from expected utility instead of Chris Starmer's paper in the Journal of Economic Literature. Citation nine is hard to find; why not use the Kahneman-Tversky Econometrica 1979 paper? It's strange that Polonsky et al.'s 2018 paper isn't published yet but it seems like a crucial citation anyway.

The phrase "it is necessary to investigate how the data sets and models interact" needs more explanation. What do you mean by interaction? A preview would be helpful if you plan on showing this interaction.

Lines 51-53 suggest that having many parameters lowers overfitting risk, which I don't agree with unless there's some clarification for your reasoning. You cite (26) what does that paper say that is convincing on this point?

Figure 1 looks great but needs more explanatory captions, especially for (d) NeuralNetworks and (d) evaluation graph sections. What does the lower right pink/blue higher/lower mean? In general, your figure captions should be longer to make Fig 1 more self-contained.

Define terms like fine-tuning, expressiveness, and pre-training as they might not be familiar to general readers or those knowledgeable about machine learning but unfamiliar with these specifics.

In the Table 1 caption, tell us what we're supposed to see here - transfer testing results perhaps? The same goes for Table 2: clarify that it's linear regression results between different feature sets. Explain how MSE and R-squared relate in NN fine-tuned on CPC versus choices 13K? I read this multiple times and it is hard to understand.

Figure captions should be self-contained so readers know what they're looking at without needing extra context. Figure 3 has excellent data but lacks clear narration in its captions; explains what readers should observe rather than just stating plot details.

Based on the text language, Fig 3 suggests differences in predictions between CPC-15 dataset (with large spikes at dominated/dominant strategies) & choices-13k analyses (smaller distributions for those same lotteries). This implies choice-13k models aren't accurately reflecting dominant relations, consistent with high dominance violation rates & guessing rates seen later.

Am glad to see you introduce 2nd and 3rd-order SD (pp 5-6) as these can be important theoretically but are previously only known in subtle economic applications.

Line 692 “Because the datasets considered here only contain the proportion of participants choosing one option in a binary decision, modeling, e.g., of individual differences or sequential effects, is limited.” This is a major limitation of the Peterson et al desire to sample a large number of problems but present only a small number per person. Just a comment, nothing for you to revise here.

Second comments batch about rebuttal ‘mediation’: Here are my thoughts about some of the “signed commenter” comments and Thomas et al’s rebuttals.

Language: As noted above, I agree with the previous Commenters who thought that some of the language in this Thomas et al paper is rejecting a claim that has rarely been made.

The authors of this paper quote Bhatia and He, who wrote in their comment on Peterson et al that “Instead of relying on the intuitions and (potentially limited) intellect of human researchers, the task of theory generation can be outsourced to powerful machine-learning algorithms”. Taking a cue from this Bhatia-He quote, the authors’ own language (in the abstract) that “...theory generation still cannot be outsourced entirely to powerful machine learning black boxes”.

As the signed Commenter notes (in the rebuttal material I received), Peterson et al never make such a bold “outsourced entirely” claim in print and they clearly do not agree with it.

For example, Peterson (p 1213) wrote that “Human ingenuity will also be required for potentially translating this descriptive theory [MOT] into normative and process models (38, 39)”

Dataset bias (a/k/a transfer performance)

P 8 Signed comment says: “We were also concerned about transfer performance, and to address this Peterson et al. (2021) presented a second new dataset obtained using a different method (with more problems per participant) and population from choices13k. Like cpc15, this dataset is also meaningfully different from choices13k ($r=0.80$), yet when models trained on choices13k were tested on this new one, our central results were replicated”

First, Thomas et al are commenting on what was reported in Peterson et al. In the SOM of that Peterson et al paper, it says (p 18) “To allay concerns about data quality and platform-specific bias...” as a justification for getting more data. In reading what was written, the term dataset bias and transfer are not used except for concern about “platform-specific bias” (Mturk vs Prolific). This is an extremely minor source of bias when you think about all the 8 billion people in the world that could be sampled.

I also do not fully agree with the commenter’s assertion that “this dataset is also meaningfully different

from choices 13k ($r=.80$)..." While $r=.80$ is not 1, choices13k are only based on 15 choices per problem. So I suspect (but haven't computed) that there is likely to be enough noise to make $r=.80$ even if the two populations were using exactly the same underlying choice process, just due to sampling error. So I question whether $r=.80$ is an actual index of "meaningfully different" (leaving aside what "meaningfully" means). Furthermore, Prolific vs Mturk, and 60 (most-responded-to) gambles instead of 20, and identical gambles, are not substantial changes.

P 10 Signed comment said "As a result of their analysis, Thomas et al. claim that choices13k is biased, implying that others are not. We take the view that all datasets are biased. The main impetus for our work was to avoid what we view as the worst kind of bias: that introduced by only looking at a small portion of human behavior."

Once again, the commenter is not describing what was written in the paper, but is adding additional commentary (which is certainly part of what's useful about peer-review). Of course, all datasets are biased. That is neither adding new information nor successfully criticizing Thomas et al's desire to study dataset bias in your application. The commenter is being too defensive and unconstructive by suggesting that "...[the] claim that choices13k is biased, implying that others are not". There is no such "implied" claim in Thomas et al's paper and I am sure they do not agree with it. Stick to constructive criticism of what they actually said rather than what you imagine a reader might think was implied.

Thomas are just interested in what dataset bias exists in the sense that the NNs generate different transferability for cpc15 vs choices13k. By the way, I do think that 14% of violations of stochastic dominance in the choice13k dataset is unnaturally high (compared to many smaller-sample experiments, mostly run pre-MTurk). And the estimated 27% pure guessing rate seems a little high (higher than typical test-retest unreliability rates). Furthermore, the fact that the choice13k subjects did not have to lose money for negative outcomes is also a shocking design error in my view. There is little chance you would get the same degree of aversion to loss if it was done in a dataset protocol with actual losses. Experimental economists have been very careful to try to impose losses (typically from a prepaid or start-of-experiment stake) because whether loss aversion exists is important, and they want efforts to generate loss valuation to be lifelike.

I also disagree that "looking at a small portion of human behavior" is the "worst kind of bias". There are many kinds of dataset bias. Which is most harmful will depend on the application and its societal implication. For example, it is hard to agree with what you regard as the worst kind of dataset bias when there are well-known biases from studying genetic databases without sampling the world or using ML trained on faces or medical data from only one race or gender.

SCOPE The implicit assertion of "signed comment" is that you have avoided the "worst kind of bias" by expanding coverage of binary choice gamble features. From the point of view of social scientists who

have studied financial risk-taking broadly, your study of binary risky choices over very small money amounts, with one certain choice and another gamble (and no payment for loss), is a tiny portion of the entire space of financial risks people choose among in their lives. The expansion to 13k problems is large but only within a narrow portion of the large space of how to measure risk taking. You have expanded coverage from one small portion of a vast space to a slightly larger small portion.

Here are some examples of other studies which sampled differently:

- Many studies use “budget lines” in which people allocate points from a budget to possible outcomes. This is essentially an infinite set of choices (not just two) which are highly constrained within a budget line set, but vary across different budget lines. (cites Choi, Syngjoo, Raymond Fisman, Douglas M. Gale, and Shachar Kariv. 2007. "Revealing Preferences Graphically: An Old Method Gets a New Tool Kit." *American Economic Review*, 97 (2): 153-158).
- Various groups used risky distributions with many outcomes more like lottery tickets with multiple prize levels. See Lola L Lopes, Gregg C Oden, The Role of Aspiration Level in Risky Choice: A Comparison of Cumulative Prospect Theory and SP/A Theory, *Journal of Mathematical Psychology*, 43(2) 1999, 286-313,. Also Goldstein, DanielG.; Eric J. Johnson; William F. Sharpe (2008). "Choosing Outcomes Versus Choosing Products: Consumer-Focused Retirement Investment Advice". *Journal of Consumer Research*. 35 (3): 440–456.
- Economists, particularly, are often concerned about dataset bias generalizing from small money amounts to large. One approach is to sample people from countries that are literate but poor (so that typical Western research budgets have a lot of purchasing power for the sampled people, on the order of 20x or so of your stakes). (cites Kachelmeier, Steven J and Shehata, Mohamed, (1992), Examining Risk Preferences Under High Monetary Incentives: Experimental Evidence from the People's Republic of China, *American Economic Review*, 82, issue 5, p. 1120-41; Tanaka, Tomomi, Colin F. Camerer, and Quang Nguyen. 2010. "Risk and Time Preferences: Linking Experimental and Household Survey Data from Vietnam." *American Economic Review*, 100 (1): 557-71; and many others)
- Many studies do not compare binary choices at all but instead elicit certain money-equivalents for individual gamble (auctions for risky goods, including treasury bonds and some kinds of stock auctions, are of this kind). Context-dependent models which compare features of two lotteries in binary choices are likely to generalize extremely badly to this setting because there is no “context” in valuing a single example until the valuation is given. An example is Lattimore et al (1989), who elicit certainty equivalents, and also include subjects who are college students and also prisoners convicted of property crimes https://www.nber.org/system/files/working_papers/t0081/t0081.pdf

P 6, Signed comment says “We first address (a). Thomas et al. observe that the noise in our data (at the level of choice proportions per decision problem) is higher than cpc15. This is to be expected, as we obtained choices from fewer participants per problem than cpc15 (i.e., ~15 vs. >100), as reported in our work.”

This comment is correct but it still illustrates a particular purely statistical kind of dataset “bias”. Sampling only 20 problems per participant, and 15 responses per problem, is just too low a sample. If you are then using statistics such as the percentage of people choosing A rather than B, you only have 16 possible values of that statistic. That is coarse and cannot be doing you any favors in trying to generalize.

P 11 Signed Comment says “There are significant issues with the way they frame this analysis as a response to claims we wouldn’t endorse, the specific comparisons they present between models maximizing the apparent similarity between those models, and the possible interpretation of their conclusions about noise, transfer performance, and bias being a result of a lack of care in our own analyses, rather than intrinsic consequences of the statistical structure of the task that we worked hard to mitigate”

Once again, this is just defensive and requires reading your mind (rather than your paper) to understand. As noted, I agree about the “response to claims we wouldn’t endorse” but this just requires a title change and reframing to the focus on dataset bias. That is not a sufficient reason to reject the paper. As far as I could tell, Thomas et al never say or even insinuate that there is a “lack of care in [y]our own analyses”. You may be reasonably concerned that when this paper is published some readers, or more likely nonreaders, will conclude that your paper had a “lack of care”. If you are worried about that, your Commenter’s main job is to say precisely what the “significant issues” are—in the specific language of their paper—that might lead a reader to the “possible interpretation” about your “lack of care”. If you think there are sentences in Thomas et al that suggest that interpretation, then say specifically how you would like them to rephrase what you think can lead to the “possible interpretation” you fear. No author can be expected to reply to a comment like this by reading your mind about how to change their own text.

Finally, I don’t see any evidence in your paper or your comments about how you “worked hard to mitigate...intrinsic consequences of the statistical structure of the task” (am rearranging your language here). This is an interesting point but for this type of back-and-forth you need to say more about how exactly you “worked hard...”. You may well have but I don’t understand what exactly is meant by your phrasing.

It is unfortunate that the authors could not access the Prolific sample. The Peterson (2021) published Science paper states at the end “All data are available to the public without registration at <https://github.com/jcpeterson/choices13k>.”; This should be remedied quickly.

Author Rebuttal, first revision:

Reviewer #1 (Remarks to the Author):

The revised draft addresses my main concerns with the original manuscript. I think that the paper highlights an interesting observation. The existence of large differences between the decisions people make in different experiments (the data set bias), suggests that running large experiments does not guarantee good generalization of the results to different settings. To facilitate generalization (and useful predictions) it is important to clarify the difference between the different data sets. The paper shows that it is not easy to achieve this goal with traditional ML methods, and demonstrates a natural theoretic analysis can help.

The paper could be improved by adding analyses that examine the value of the theoretical idea it proposes. In theory, it is possible that the explanatory value of the added noise hypothesis is only a good post-hoc explanation of the current findings. Yet, I think that this analysis can wait to future research.

A minor comment: Figure 1 should be updated to include the CPC18 analyses

Thank you for your updated assessment of our manuscript.

We agree that our noise hypothesis might be a post-hoc explanation, yet we are unsure which analysis you refer to that we could add so as to validate our results in the current study. Importantly, that the added structured noise in our hybrid model results in a better fitting of the data on the choices13k dataset is a strong indication that this model indeed captures meaningful aspects of the difference in participants' behavior across studies and datasets. Further support for such an interpretation of these results comes from other investigations into datasets collected over Amazon Mechanical Turk, which we cite and discuss in the discussion section. Future research, ideally with individual participants' instead of aggregated data, should investigate this question further.

Figure 1 currently only shows the datasets we have trained on. In the first draft they were identical with the ones we have evaluated on, but now we also evaluate models on CPC18. We are not sure how to indicate another dataset that is not used for training but only for evaluation. To avoid visual clutter and confusion, we have decided not to include CPC18 in Figure 1.

However, we have made several smaller changes to subfigure e) illustrating the evaluation of neural networks.

Reviewer #3 (Remarks to the Author):

Review of "Still no free lunch from deep learning: Data, models, and theory of human economic decision-making" NHB

This paper reanalyzes several aspects of prominently published work by Peterson et al (2021, Science) building on a very similar 2019 Bourgin paper.

I agree with the previous Commenters who thought that some of the language in this paper-- including the "no free lunch" in the title, which the authors must get rid of-- is rejecting a claim that has rarely been made.

Instead, the main contribution of this paper is to dig into how much dataset bias there might be in Peterson et al, and what causes it. This is a very interesting question because "dataset bias" is a well-known challenge in learning from datasets but is not often recognized by that name in experimental social science.

This comment is therefore useful in extending scientists' understanding of the predictive scope of this paper and how to explore its limits, empirically.

The dataset bias referred to is whether an NN or other theoretical extrapolation from one dataset (say choice13k in this case) generalizes well in predicting an unseen different dataset.

The value of machine learning-enhanced theory discovery presumably depends, as do all other kinds of theory generation, on dataset bias not being too large and unpredictable. (If dataset bias is large, that is manageable as long as there is some theory or guidelines about when to expect poor generalization.) So the question of dataset bias is absolutely central to the general point Peterson et al were making about the usefulness of the large data set approach. However, it is also clear that dataset bias clearly was not the focus of Peterson et al's paper at all (there term is not even mentioned in their paper).

However, concern for dataset bias is important in thinking about what ideas derived from the analysis of specific data sets would generalize to very different kinds of data sets used to test risky choice modeling. Many of the Thomas et al authors and the original

Peterson et al authors may not be aware that in the larger space of possible experimental (and natural) choices involving financial risk, the dataset differences between cpc15, cpc18, and choice13k are small. They all use binary choices between two money gambles with small amounts of money, and not too many outcomes, which are presented as descriptions. Subjects choose one choice in a pair and can't express a strength of preference or indifference. Even within the large domain of financial risk experiments, and risky choices in everyday life, there is hardly any daylight between the cpc sets and choice13k.

*So if there is dataset bias here, as measured by the low transferability of predictions derived from one of these datasets to another, that is surely a lower bound on how bad the data set bias might be for very different tasks, amounts of money, and very likely subject populations. I will discuss this further below at location *SCOPE**

You should think about a retitling such as “An investigation into dataset bias in machine-learned theory discovery”

Thank you for the overall positive evaluation as well as constructive feedback of our manuscript.

We agree that the citation about automatic theory discovery is not by the authors of the original paper. Their description is much more measured: “The use of large datasets has revolutionized machine learning, computer vision, and artificial intelligence. Our study is one of the first to use a similar methodology in systematically investigating theories of human cognition. We believe that the use of large datasets coupled with machine-learning algorithms offers enormous potential for uncovering new cognitive and behavioral phenomena that would be difficult to identify without such tools”. Indeed, it is the commentary published along the original article which puts it more boldly: “Instead of relying on the intuitions and (potentially limited) intellect of human researchers, the task of theory generation can be outsourced to powerful machine-learning algorithms”.

While we accept your critique about our usage of language, we think that the proposed title “*An investigation into dataset bias in machine-learned theory discovery*” does not capture a large part of the current study, which, as you say, does not only “dig into how much dataset bias there might be in Peterson et al.” but also “*what causes it*”. Therefore, we would like to suggest the following title: “Fitting the noise: dataset bias in supervised machine learned theories of economic decisions”.

Regarding the point of the “*space of possible experimental (and natural) choices*”, we could not agree more. This is indeed the reason why we included and discussed recent developments

advocating richer decision-making tasks and data set together with appropriate modeling in the discussion section. We additionally discuss this point later in your scope section.

I am seeing this ms. for the first time along with the authors' rebuttal letter that includes original referee report questions they tried to answer. My comments are therefore in two groups.

The first batch is specific comments about improving the paper, which is mostly editorial. The second batch of comments is my attempt to 'mediate' between the "signed commenter" comments and the authors' responses.

First comments batch about the Thomas et al paper.

Some citations are a bit odd. For instance, citation one cites Glimcher on deviations from expected utility instead of Chris Starmer's paper in the Journal of Economic Literature. Citation nine is hard to find; why not use the Kahneman-Tversky Econometrica 1979 paper? It's strange that Polonsky et al.'s 2018 paper isn't published yet but it seems like a crucial citation anyway.

Thank you for the suggested citations. We are very happy to include all of them and have updated the references accordingly. Since the Announcement of the Choice Prediction Competition 2018 (CPC18) is not a published paper, we have decided to cite the arXiv preprint from a year later, where the organizers of the competition discuss the dataset as well as the results.

The phrase "it is necessary to investigate how the data sets and models interact" needs more explanation. What do you mean by interaction? A preview would be helpful if you plan on showing this interaction.

Sorry for not being clearer. What we want to express here is that models and datasets are not independent. Having two models, A and B, and two datasets, XA and XB, which are generated from models A and B, respectively, does not necessarily result in A always better fitting XA and B always better fitting XB. This may depend on the size of the datasets and the complexity of the models. And, these can interact in that smaller datasets are always better fit by the less complex model whereas larger datasets from the more complex model are only fit well by the generating, complex model. A model that performs best on one type of dataset does not necessarily transfer to different datasets. For example, the effect of model complexity on

performance might depend on the specifics of a data set. This is what we mean by “interaction between data sets and models”. Here, adding decision noise increases performance on choices13k, but decreases performance on previous laboratory datasets. Another interaction is that the human behavior in the CPC15 dataset might be well fit by the Peterson et al. NN, but we cannot find this out based on the dataset, which is too small for proper training.

We have added some clarifications at this point in the introduction.

Lines 51-53 suggest that having many parameters lowers overfitting risk, which I don't agree with unless there's some clarification for your reasoning. You cite (26) what does that paper say that is convincing on this point?

Sorry for not being clearer. We are referring to the “double descent” phenomenon in machine learning, which has gained prominence in the last few years in the context of deep neural networks. When increasing the number of parameters for machine learning models, the test error first decreases, then increases, but then decreases again. From the classical perspective of the bias-variance tradeoff, this second decrease is unexpected, because models with higher complexity should overfit the training set and perform worse on the test set. However, the double descent phenomenon is typical for highly overparameterized models such as neural networks. The cited paper (26, now 27) reconciles these seemingly conflicting viewpoints by arguing that once the model complexity is high enough that it contains functions with the appropriate inductive biases for the learning problem at hand, the learned model will tend towards those solutions.

We tried to make our argument more clear at that point in the manuscript, and have referred to more literature discussing the phenomenon of double descent in high dimensional NNs.

Figure 1 looks great but needs more explanatory captions, especially for (d) NeuralNetworks and (d) evaluation graph sections. What does the lower right pink/blue higher/lower mean? In general, your figure captions should be longer to make Fig 1 more self-contained.

Yes, thank you for pointing this out. The lower right shows an example of feature importance values that the XAI method SHAP returns for an exemplary gamble. We have added this and more information to make Figure 1 more self-contained.

Define terms like fine-tuning, expressiveness, and pre-training as they might not be familiar to general readers or those knowledgeable about machine learning but unfamiliar with these specifics.

Thank you for this suggestion. We added short explanations for these terms.

In the Table 1 caption, tell us what we're supposed to see here - transfer testing results perhaps? The same goes for Table 2: clarify that it's linear regression results between different feature sets. Explain how MSE and R-squared relate in NN fine-tuned on CPC versus choices 13K? I read this multiple times and it is hard to understand.

Figure captions should be self-contained so readers know what they're looking at without needing extra context. Figure 3 has excellent data but lacks clear narration in its captions; explains what readers should observe rather than just stating plot details.

Sorry for the lack of clarity. We have added subtitles to multiple figures and have added more context to the captions of all figures and tables, making them more self-explanatory.

Based on the text language, Fig 3 suggests differences in predictions between CPC-15 dataset (with large spikes at dominated/dominant strategies) & choices-13k analyses (smaller distributions for those same lotteries). This implies choice-13k models aren't accurately reflecting dominant relations, consistent with high dominance violation rates & guessing rates seen later.

Am glad to see you introduce 2nd and 3rd-order SD (pp 5-6) as these can be important theoretically but are previously only known in subtle economic applications.

Line 692 "Because the datasets considered here only contain the proportion of participants choosing one option in a binary decision, modeling, e.g., of individual differences or sequential effects, is limited." This is a major limitation of the Peterson et al desire to sample a large number of problems but present only a small number per person. Just a comment, nothing for you to revise here.

Second comments batch about rebuttal ‘mediation’: Here are my thoughts about some of the “signed commenter” comments and Thomas et al’s rebuttals.

Language: As noted above, I agree with the previous Commenters who thought that some of the language in this Thomas et al paper is rejecting a claim that has rarely been made.

The authors of this paper quote Bhatia and He, who wrote in their comment on Peterson et al that “Instead of relying on the intuitions and (potentially limited) intellect of human researchers, the task of theory generation can be outsourced to powerful machine-learning algorithms”. Taking a cue from this Bhatia-He quote, the authors’ own language (in the abstract) that “...theory generation still cannot be outsourced entirely to powerful machine learning black boxes”.

As the signed Commenter notes (in the rebuttal material I received), Peterson et al never make such a bold “outsourced entirely” claim in print and they clearly do not agree with it.

For example, Peterson (p 1213) wrote that “Human ingenuity will also be required for potentially translating this descriptive theory [MOT] into normative and process models (38, 39)”

We agree that some claims of fully automated theory discovery are not supported by the authors of the original paper, and have in consequence updated a few sentences in the introduction and discussion to reflect this.

Dataset bias (a/k/a transfer performance)

P 8 Signed comment says: “We were also concerned about transfer performance, and to address this Peterson et al. (2021) presented a second new dataset obtained using a different method (with more problems per participant) and population from choices13k. Like cpc15, this dataset is also meaningfully different from choices13k ($r=0.80$), yet when models trained on choices13k were tested on this new one, our central results were replicated”

First, Thomas et al are commenting on what was reported in Peterson et al. In the SOM of that Peterson et al paper, it says (p 18) “To allay concerns about data quality and platform-specific bias...” as a justification for getting more data. In reading what was written, the term dataset bias and transfer are not used except for concern about “platform-specific bias” (Mturk vs Prolific). This is an extremely minor source of bias when you think about all the 8 billion people in the world that could be sampled.

I also do not fully agree with the commenter’s assertion that “this dataset is also meaningfully different from choices 13k ($r=.80$)...” While $r=.80$ is not 1, choices13k are only based on 15 choices per problem. So I suspect (but haven’t computed) that there is likely to be enough noise to make $r=.80$ even if the two populations were using exactly the same underlying choice process, just due to sampling error. So I question whether $r=.80$ is an actual index of “meaningfully different” (leaving aside what “meaningfully” means). Furthermore, Prolific vs Mturk, and 60 (most-responded-to) gambles instead of 20, and identical gambles, are not substantial changes.

P 10 Signed comment said “As a result of their analysis, Thomas et al. claim that choices13k is biased, implying that others are not. We take the view that all datasets are biased. The main impetus for our work was to avoid what we view as the worst kind of bias: that introduced by only looking at a small portion of human behavior.”

Once again, the commenter is not describing what was written in the paper, but is adding additional commentary (which is certainly part of what’s useful about peer-review). Of course, all datasets are biased. That is neither adding new information nor successfully criticizing Thomas et al’s desire to study dataset bias in your application. The commenter is being too defensive and unconstructive by suggesting that “...[the] claim that choices13k is biased, implying that others are not”. There is no such “implied” claim in Thomas et al’s paper and I am sure they do not agree with it. Stick to constructive criticism of what they actually said rather than what you imagine a reader might think was implied.

Thomas are just interested in what dataset bias exists in the sense that the NNs generate different transferability for cpc15 vs choices13k. By the way, I do think that 14% of violations of stochastic dominance in the choice13k dataset is unnaturally high (compared to many smaller-sample experiments, mostly run pre-MTurk). And the estimated 27% pure guessing rate seems a little high (higher than typical test-retest unreliability rates). Furthermore, the fact that the choice13k subjects did not have to lose money for negative outcomes is also a shocking design error in my view. There is little

chance you would get the same degree of aversion to loss if it was done in a dataset protocol with actual losses. Experimental economists have been very careful to try to impose losses (typically from a prepaid or start-of-experiment stake) because whether loss aversion exists is important, and they want efforts to generate loss valuation to be lifelike.

I also disagree that “looking at a small portion of human behavior” is the “worst kind of bias”. There are many kinds of dataset bias. Which is most harmful will depend on the application and its societal implication. For example, it is hard to agree with what you regard as the worst kind of dataset bias when there are well-known biases from studying genetic databases without sampling the world or using ML trained on faces or medical data from only one race or gender.

SCOPE The implicit assertion of "signed comment" is that you have avoided the “worst kind of bias” by expanding coverage of binary choice gamble features. From the point of view of social scientists who have studied financial risk-taking broadly, your study of binary risky choices over very small money amounts, with one certain choice and another gamble (and no payment for loss), is a tiny portion of the entire space of financial risks people choose among in their lives. The expansion to 13k problems is large but only within a narrow portion of the large space of how to measure risk taking. You have expanded coverage from one small portion of a vast space to a slightly larger small portion.

Here are some examples of other studies which sampled differently:

- Many studies use “budget lines” in which people allocate points from a budget to possible outcomes. This is essentially an infinite set of choices (not just two) which are highly constrained within a budget line set, but vary across different budget lines. (cites Choi, Syngjoo, Raymond Fisman, Douglas M. Gale, and Shachar Kariv. 2007. "Revealing Preferences Graphically: An Old Method Gets a New Tool Kit." *American Economic Review*, 97 (2): 153-158).
- Various groups used risky distributions with many outcomes more like lottery tickets with multiple prize levels. See Lola L Lopes, Gregg C Oden, *The Role of Aspiration Level in Risky Choice: A Comparison of Cumulative Prospect Theory and SP/A Theory*, *Journal of Mathematical Psychology*, 43(2) 1999, 286-313,). Also Goldstein, DanielG.; Eric J. Johnson; William F. Sharpe (2008). "Choosing Outcomes Versus Choosing Products: Consumer-Focused Retirement Investment Advice". *Journal of Consumer Research*. 35 (3): 440–456.

- *Economists, particularly, are often concerned about dataset bias generalizing from small money amounts to large. One approach is to sample people from countries that are literate but poor (so that typical Western research budgets have a lot of purchasing power for the sampled people, on the order of 20x or so of your stakes). (cites Kachelmeier, Steven J and Shehata, Mohamed, (1992), Examining Risk Preferences Under High Monetary Incentives: Experimental Evidence from the People's Republic of China, American Economic Review, 82, issue 5, p. 1120-41; Tanaka, Tomomi, Colin F. Camerer, and Quang Nguyen. 2010. "Risk and Time Preferences: Linking Experimental and Household Survey Data from Vietnam." American Economic Review, 100 (1): 557-71; and many others)*
- *Many studies do not compare binary choices at all but instead elicit certain money-equivalents for individual gamble (auctions for risky goods, including treasury bonds and some kinds of stock auctions, are of this kind). Context-dependent models which compare features of two lotteries in binary choices are likely to generalize extremely badly to this setting because there is no "context" in valuing a single example until the valuation is given. An example is Lattimore et al (1989), who elicit certainty equivalents, and also include subjects who are college students and also prisoners convicted of property crimes*
https://www.nber.org/system/files/working_papers/t0081/t0081.pdf

We fully agree that the setting of the CPC15, CPC18, and choices13k experiments is far from natural. Indeed, our lab has particularly focused over the last decades on finding computational models of sequential decision-making for more natural behavior, often involving the body. We explicitly refer to this work and the current renaissance of interest in naturalistic tasks in neuroscience and behavioral science in the discussion section. Thus, we are very grateful for the discussion around the scope and the additional references, and have added some of it to our discussion.

P 6, Signed comment says "We first address (a). Thomas et al. observe that the noise in our data (at the level of choice proportions per decision problem) is higher than cpc15. This is to be expected, as we obtained choices from fewer participants per problem than cpc15 (i.e., ~15 vs. >100), as reported in our work."

This comment is correct but it still illustrates a particular purely statistical kind of dataset "bias". Sampling only 20 problems per participant, and 15 responses per problem, is just too low a sample. If you are then using statistics such as the percentage of people choosing A rather than B, you only have 16 possible values of that statistic. That is coarse and cannot be doing you any favors in trying to generalize.

This is exactly right. Our analysis using the probabilistic generative model shows that the noise is not merely a result of the number of participants, as we explicitly correct for it by including the number of responses n in the Bayesian generative model. And indeed, this analysis shows that there is additional noise per participant to explain the data in choices13k.

P 11 Signed Comment says “There are significant issues with the way they frame this analysis as a response to claims we wouldn’t endorse, the specific comparisons they present between models maximizing the apparent similarity between those models, and the possible interpretation of their conclusions about noise, transfer performance, and bias being a result of a lack of care in our own analyses, rather than intrinsic consequences of the statistical structure of the task that we worked hard to mitigate”

Once again, this is just defensive and requires reading your mind (rather than your paper) to understand. As noted, I agree about the “response to claims we wouldn’t endorse” but this just requires a title change and reframing to the focus on dataset bias. That is not a sufficient reason to reject the paper. As far as I could tell, Thomas et al never say or even insinuate that there is a “lack of care in [y]our own analyses”. You may be reasonably concerned that when this paper is published some readers, or more likely nonreaders, will conclude that your paper had a “lack of care”. If you are worried about that, your Commenter’s main job is to say precisely what the “significant issues” are—in the specific language of their paper—that might lead a reader to the “possible interpretation” about your “lack of care”. If you think there are sentences in Thomas et al that suggest that interpretation, then say specifically how you would like them to rephrase what you think can lead to the “possible interpretation” you fear. No author can be expected to reply to a comment like this by reading your mind about how to change their own text.

To avoid any such impression, we have added language in the description of the Peterson et al. study throughout the paper that hopefully reinforces that indeed we do not insinuate a lack of care on the part of the authors.

Finally, I don’t see any evidence in your paper or your comments about how you “worked hard to mitigate...intrinsic consequences of the statistical structure of the task” (am rearranging your language here). This is an interesting point but for this type of back-and-forth you need to say more about how exactly you “worked hard...”. You may well have but I don’t understand what exactly is meant by your phrasing.

It is unfortunate that the authors could not access the Prolific sample. The Peterson (2021) published Science paper states at the end "All data are available to the public without registration at <https://github.com/jcpeterson/choices13k>; This should be remedied quickly.

Overall, we would like to thank reviewer 3 for their constructive and substantial feedback, as well as for mediating between our manuscript and the signed comment.

Final Decision Letter:

Dear Mr Thomas,

We are pleased to inform you that your Article "Modeling dataset bias in machine-learned theories of economic decision making", has now been accepted for publication in Nature Human Behaviour.

Please note that *Nature Human Behaviour* is a Transformative Journal (TJ). Authors may publish their research with us through the traditional subscription access route or make their paper immediately open access through payment of an article-processing charge (APC). Authors will not be required to make a final decision about access to their article until it has been accepted. Find out more about Transformative Journals

With best regards,

Jamie

Dr Jamie Horder
Senior Editor
Nature Human Behaviour